# Programmable nano-reactors for stochastic sensing

Wendong Jia [1,2], Chengzhen Hu [1,2], Yuqin Wang [1,2], Yuming Gu[1,2], Guangrui Qian[3], Xiaoyu Du[1,2], Liying Wang[1,2], Yao Liu [1,2], Jiao Cao [1,2], Shanyu Zhang[1,2], Shuanghong Yan [1,2], Panke Zhang [1], Jing Ma [1], Hong-Yuan Chen [1] & Shuo Huang [1,2 ✉]

Chemical reactions of single molecules, caused by rapid formation or breaking of chemical bonds, are difficult to observe even with state-of-the-art instruments. A biological nanopore can be engineered into a single molecule reactor, capable of detecting the binding of a monatomic ion or the transient appearance of chemical intermediates. Pore engineering of this type is however technically challenging, which has significantly restricted further development of this technique. We propose a versatile strategy, "programmable nano-reactors for stochastic sensing" (PNRSS), by which a variety of single molecule reactions of hydrogen peroxide, metal ions, ethylene glycol, glycerol, lactic acid, vitamins, catecholamines or nucleoside analogues can be observed directly. PNRSS presents a refined sensing resolution which can be further enhanced by an artificial intelligence algorithm. Remdesivir, a nucleoside analogue and an investigational anti-viral drug used to treat COVID-19, can be distinguished from its active triphosphate form by PNRSS, suggesting applications in pharmacokinetics or drug screening.

[1] State Key Laboratory of Analytical Chemistry for Life Sciences, School of Chemistry and Chemical Engineering, Nanjing University, 210023 Nanjing, China. [2] Chemistry and Biomedicine Innovation Center (ChemBIC), Nanjing University, 210023 Nanjing, China. [3] Intelligence Qubic Technology Co. Ltd, Beijing, China. ✉email: shuo.huang@nju.edu.cn

Although described and formulated in terms of single molecules, chemical reactions are rarely monitored or characterized at the single-molecule level but in ensembles. Few state-of-the-art instruments, including scanning probe microscopy[1], tip/surface-enhanced Raman spectroscopy[2], molecular junctions[3], single-molecule force spectroscopy[4] or biological nanopores[5], are capable of resolving discrete steps of the chemical reactions of a single molecule. These state-of-the-art techniques, however, report different aspects of single-molecule properties. Scanning tunnelling microscopy and molecular junctions focus on the electronic state of the target molecule. Raman spectroscopy focuses on investigation of the vibration of chemical bonds, whereas force spectroscopy measures the elastic properties molecules.

Biological nanopores, including α-haemolysin (α-HL)[6], *Mycobacterium smegmatis* porin A (MspA)[7], aerolysine[8], CsgG[9], cytolysin A (ClyA)[10], fragaceatoxin C (FraC)[11], pleurotolysin (PlyA/B)[12], outer membrane protein G[13], phi29 connector[14] and a few others, are a category of large channel proteins developed for single-molecule sensing. Conformational properties of nucleic acids[15], peptides[16], proteins[17] and small molecules[18] can be probed during translocation of the analyte through the pore constriction. The nanopore sequencer MinION[TM], which applies a similar measurement scheme, is able to directly report nucleic acid sequences[19]. Pioneered by Bayley et al.[5], an engineered α-HL nanopore, which, with a sole, fixed reactive site within its lumen, is capable of reacting with discrete freely translocating reactants, forming a single-molecule reactor, capable to resolve binding of a single metal ion[20–22]. However, nanopore single-molecule chemistry measurements performed by α-HL generally report a weak event amplitude, measuring 1–5 pA[23], which prohibits it from gaining a further refined resolution. Undesired reactive sites evenly distributed within the cylindrical lumen of α-HL may also interfere with the measurement, requiring excessive pore engineering efforts[22]. Most biological nanopores show an oligomeric symmetry[6] and introduction of a sole reactive site therefore requires a significant effort to produce a hetero-oligomeric assembly[5], a niche technique mastered by only few in the field. Nanopore single-molecule chemistry measurements performed by engineered homo-oligomeric porins will inevitably report simultaneous binding from multiple reactants, not suitable for event recognition and quantification[24–27]. Modifications to the pore lumen may interfere unpredictably with the assembly or the stability of the pore and, consequently, a considerable effort of protein screening cannot therefore be avoided. To introduce unnatural reactive sites into the pore lumen, a semi-synthetic α-HL was prepared by native chemical ligation (NCL)[28,29]. However, nanopore preparation by NCL requires extremely complicated purification procedures and is not always guaranteed to work when engineering to a different site or pore type is to be performed. Some other state-of-the-art techniques apply internal adaptors such as cyclodextrins[30,31] or proteins[18,32] inside the pore lumen to acquire new sensing abilities. These approaches require the existence of suitable adaptors. Accommodation of large enzymatic adaptors may also report an undesired resolution to discriminate molecular analytes with extremely minor structural differences. A technical breakthrough to achieve a complete freedom in the placement of any type, number or spatial combination of reactive sites to any spot of the pore lumen is required to resolve these problems.

In this work, we present a versatile strategy, 'programmable nano-reactors for stochastic sensing' (PNRSS) to democratize nanopore-based single-molecule chemistry investigations. PNRSS, tentatively pronounced as /p'na:s/, is carried out cooperatively with a strand of synthetic polymer and a nanopore, defined respectively as the PNRSS strand and the PNRSS pore. A PNRSS strand is itself composed of functional modules defined as the tether site, the extension section, the reaction section and the traction section (Fig. 1a). This configuration has much lowered the technical hurdle of pore engineering. Design of PNRSS strand is fully flexible and the corresponding synthesis can be performed by low-cost commercial services.

## Results

Experimentally, the tether site of a PNRSS strand serves to conjugate one end of the strand to a tether such as a streptavidin. A streptavidin-tethered PNRSS strand is electrophoretically docked, remaining fully stretched in the PNRSS pore (Fig. 1b). The length of the extension section of a PNRSS strand is optimized with a 3.5 Å precision so that the reaction section is located at the pore restriction for optimum performance. This configuration has been previously applied to discriminate between different nucleic acid sequences[15,33,34] but has never been applied to perform nanopore single-molecule chemistry measurements. One or multiple reactive sites within the reaction section form a fixed reactant, which directly participates in the reaction under investigation. The traction section serves to maintain the electrophoretic force on the strand. The mobile reactant, which binds to the fixed reactant, is placed in the measurement environment and acts when bound to the fixed reactant. A PNRSS strand can be composed of any synthetic polymer such as nucleic acid, peptide, polysaccharide or combinations thereof; however, to study a wider variety of single-molecule reactions, the composition of the PNRSS strand should be arbitrarily programmable. DNA, the most investigated synthetic polymer, can be easily and economically synthesized, chemically modified, enzymatically treated, purified, characterized and stored[35]. The method is not restricted to DNA but it is an ideal component of a PNRSS strand. Some previous demonstrations of chimeric molecules trapped inside the pore lumen[36–41] may be inspiring in the design of a PNRSS strand using peptide components. However, observation of single-molecule chemical reactions has not been reported with these trapped chimeric molecules so far.

The PNRSS pore should possess a sharp and narrow restriction for a high spatial resolution, a rigid and reproducible structure for a high measurement consistency and a chemically inert pore lumen to minimize undesired reactions. Recent reports of engineered MspA in applications of single-molecule chemistry have demonstrated its structural superiority by which a significantly enlarged event amplitude (~55 pA) was reported[27]. Although not restricted to, an MspA ('Methods') is applied to demonstrate all PNRSS measurements in this study.

Natural nucleic acid bases, such as guanine, adenine or any combination thereof, can act as a coordination ligand that can bind a metal ion[42,43]. A first PNRSS strand 13G/14G (Supplementary Table 1) had two neighbouring guanines that cooperatively bind a $Ni^{2+}$ ion (Fig. 1c). To avoid interferences from other DNA bases, these guanines were surrounded by abasic residues that cannot bind metal ions. The PNRSS measurement was carried out as described in 'Methods'. This 13G/14G strand was added to the *cis* chamber with a 10 nM final concentration. With a single pore inserted and a +180 mV potential continuously applied, an open pore current $I_0$ was initially reported. Subsequently, a PNRSS strand was captured electrophoretically and reported a static residual current, $I_p$ (Supplementary Table 2). $Ni^{2+}$, serving as the mobile reactant, was then added to the *trans* chamber with a 1 mM final concentration. This immediately results in further blockage events, reaching a level defined as $I_b$. Successive appearances of events were observed as telegraphic switching between $I_p$ and $I_b$ (Fig. 1d and Supplementary Movie 1). The concentration of $Ni^{2+}$ in *trans* was adjusted

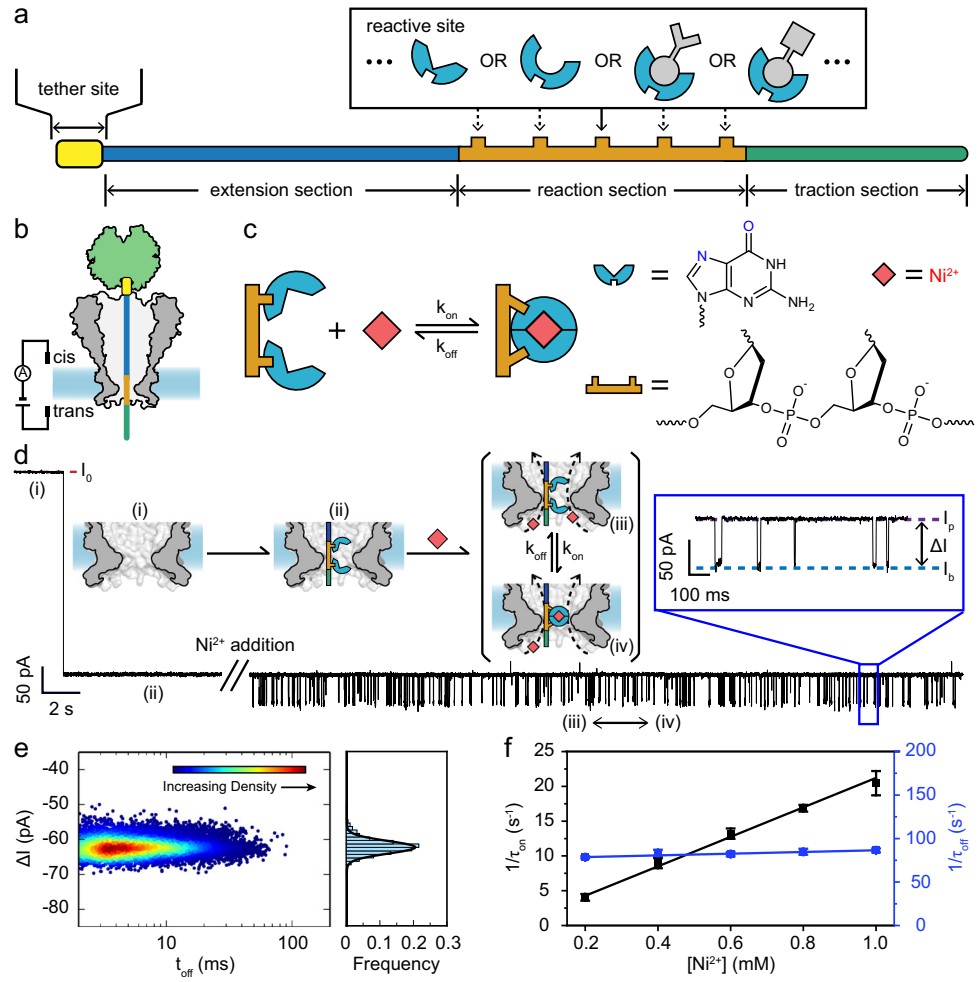

**Fig. 1 Conceptual demonstration of PNRSS. a** A PNRSS strand. A PNRSS strand is composed of functional modules as described. The reaction section, the most critical module, contains one or more reactive sites forming a fixed reactant. **b** The measurement configuration. During PNRSS, an MspA nanopore serves to dock a streptavidin-tethered PNRSS strand. The reaction section (dark yellow) is located precisely at the pore restriction for optimum performance. **c** The design of PNRSS to probe a single-molecule reaction between $Ni^{2+}$ and a dual guanine reactant. Two neighbouring guanines on the PNRSS strand cooperatively bind a $Ni^{2+}$ ion. **d** A continuous trace containing different states of a PNRSS measurement. Initially, the pore was unoccupied (i) and an open pore current was reported. A PNRSS strand was then captured by the pore, causing an immediate drop of the blockage level to $I_p$ (ii). After the placement of $Ni^{2+}$, reversible switching between $I_b$ and $I_p$ was observed, which respectively demonstrate the state when the PNRSS strand was not bound (iii) or bound (iv) with a $Ni^{2+}$. The final concentration of $Ni^{2+}$ was 0.8 mM. **e** Density scatter plot of $\triangle I$ vs. $t_{off}$. The colour scale represents the local density around each point. The density scatter plot was generated using the ggplot2 package of R. The histogram of $\triangle I$ with a Gaussian fitting is on the right of the scatter plot. Results of 13,435 events are included. **f** Concentration dependence. The reciprocal of inter-event interval ($1/\tau_{on}$) and the reciprocal of dwell time ($1/\tau_{off}$) is plotted against $[Ni^{2+}]$. The measurement was carried out as described in 'Methods'. The PNRSS strand 13G/14G (Supplementary Table 1) was applied. Nickel sulfate was added to *trans* at the desired final concentration. Error bars = SD ($N = 3$).

between 0 and 1 mM (Supplementary Fig. 1) and then the rate of event appearance clearly increased at a higher $[Ni^{2+}]$. However, these events were not observed from another PNRSS strand 14X (Supplementary Fig. 2), confirming that they originate from binding to the dual guanine ligand[43]. This phenomenon was also theoretically simulated and the N(7) and the O(6) atoms were seen to play a critical role in the coordination of a $Ni^{2+}$ (Supplementary Fig. 3, Supplementary Table 3 and 'Methods').

Core parameters that quantitatively describe any PNRSS event are summarized in Supplementary Fig. 4, in which the dwell time $t_{off}$ and the inter-event interval $t_{on}$ are defined. The blockage amplitude, $\triangle I$ is defined as $I_b - I_p$. The event scatter plot of $\triangle I$ vs. $t_{off}$ for measurement with 13G/14G and $Ni^{2+}$ demonstrates a single population of events, in which $\triangle I = \sim -60$ pA (Fig. 1e), significantly larger than that produced by a monatomic ion, which is ~2 pA in amplitude when observed with an α-HL[23]. The mean dwell time $\tau_{off}$ and the mean inter-event interval $\tau_{on}$ are

respectively derived as described in Supplementary Fig. 4. The time histograms were summarized in Supplementary Fig. 5, from which the corresponding $\tau_{on}$ and $\tau_{off}$ values were derived. As summarized in Fig. 1f, the reciprocal of the mean inter-event interval ($1/\tau_{on}$) is proportional to $[Ni^{2+}]$, consistent with a single-step bimolecular model, in which

$$1/\tau_{on} = k_{on}[Ni^{2+}] \tag{1}$$

However, the mean dwell time, $\tau_{off}$ demonstrates a negligible dependence on $[Ni^{2+}]$, consistent with a unimolecular dissociation model, in which

$$1/\tau_{off} = k_{off} \tag{2}$$

Thus, $k_{on}$ is determined as the slope of the fitted line of $1/\tau_{on}$ vs. $[Ni^{2+}]$ and $k_{off}$ is determined from the mean of the $1/\tau_{off}$ value.

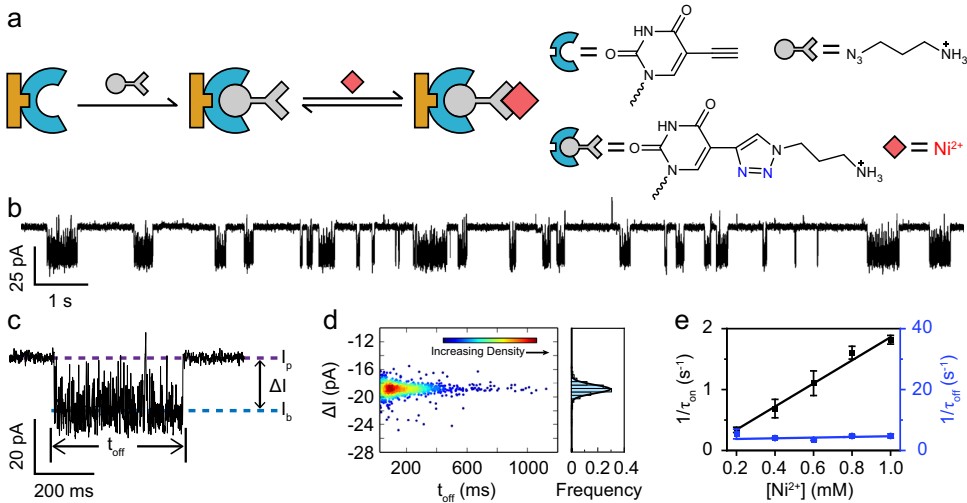

**Fig. 2 PNRSS with unnatural reactive components. a** The introduction of a 1,2,3-triazole (TAZ) to the PNRSS strand and its reaction mechanism. The PNRSS strand 14TAK (Supplementary Table 1) was reacted with 3-azidopropylamine, as described in Supplementary Fig. 11. The generated TAZ serves as the fixed reactant. $Ni^{2+}$, the mobile reactant, becomes involved in reversible coordination with a TAZ. **b** Trace demonstration. A continuous trace containing $Ni^{2+}$-binding events. Characteristic noise fluctuations were consistently observed during $Ni^{2+}$ binding. **c** A representative event. A zoomed-in demonstration of a binding event. Noise fluctuations during $Ni^{2+}$ binding indicate possible reactive intermediates observable by PNRSS. **d** Density scatter plot of $\triangle I$ vs. $t_{off}$. The colour scale represents the local density around each point. The density scatter plot was generated using the ggplot2 package of R. A highly uniform event population was observed. Results of 905 events are included. **e** Concentration dependence. The reciprocal of inter-event interval ($1/\tau_{on}$) and the reciprocal of dwell time ($1/\tau_{off}$) is plotted against $[Ni^{2+}]$ (Supplementary Table 6). The measurement was carried out as described in 'Methods'. Nickel sulfate was added to *trans* at the desired final concentration. Error bars = SD ($N = 3$).

Other divalent ions such as $Zn^{2+}$, $Cd^{2+}$, $Co^{2+}$ or $Cu^{2+}$ were also tested with 13G/14G, which demonstrates a binding preference for $Co^{2+}$ and $Cu^{2+}$ over that of $Zn^{2+}$ or $Cd^{2+}$ (Supplementary Fig. 6). Binding events from $Ni^{2+}$, $Co^{2+}$ or $Cu^{2+}$ also dramatically differ from each other, indicating that single ion discrimination is feasible by PNRSS. Different PNRSS strands such as 14A, with a sole adenine as the fixed reactant, or 14G, with guanine (Supplementary Table 1), also have provided events of $Ni^{2+}$ binding (Supplementary Figs. 7–10). The binding characteristics here are however different from those observed with 13G/14G (Supplementary Table 4), indicating that different chemical processes monitored by PNRSS were occurring. Although previously studied with nuclear magnetic resonance (NMR) or infra-red spectroscopy in ensembles[42,43], direct single-molecule observation of metal ion binding to DNA bases and their corresponding quantitative binding kinetics have not been reported before, to the best of our knowledge.

Beyond the above proof-of-concept, a much wider choice of artificial, functional DNA phosphoramadites provides a relatively unrestricted freedom in the design and synthesis of PNRSS strands, and thoroughly broadens the generality and the complexity of the downstream PNRSS measurements[44]. 5-Ethynyl-dU-CE phosphoramidite (Glen Research, USA), a thymine derivative containing an alkyne, was used in the synthesis of the PNRSS strand 14TAK (Supplementary Table 1 and 'Methods'). 14TAK contains a sole alkyne at site 14, to which any azides can be conjugated by a Huisgen copper (I)-catalysed azide-alkyne 1,3-dipolar cycloaddition (CuAAC) reaction, a widely applied click chemistry reaction[45]. 3-Azidopropylamine, one of the simplest azides, was reacted with 14TAK (Supplementary Table 1 and Supplementary Fig. 11). The product was characterized by mass spectrometry (Supplementary Fig. 11) and single-channel recording (Supplementary Fig. 12 and Supplementary Table 5). The successful conjugation was confirmed, generating a new PNRSS strand referred to as 14TAZ (Supplementary Table 1). From the reaction with CuAAC, a 1,2,3-triazole (TAZ) was generated at site 14 of 14TAZ (Fig. 2a).

It has previously been reported that the N(2) or the N(3) atom of a TAZ can serve as an electron lone pair donor and act as a coordination site to bind metal ions[46]. As a demonstration of the behaviour of a single molecule, a PNRSS measurement was carried out, in which the TAZ of the 14TAZ was applied as the fixed reactant. $Ni^{2+}$, serving as a mobile reactant, was added to *trans* with a 1 mM final concentration. With a continuously applied +180 mV potential, successive appearance of binding events with a highly characteristic noise signal was observed (Fig. 2b, c and Supplementary Movie 2). A single distribution of events was reported in the scatter plot of $\triangle I$ vs. $t_{off}$ (Fig. 2d). The rate of event appearance was clearly increased by upregulating the $Ni^{2+}$ concentration in *trans* from 0 to 1 mM (Supplementary Figs. 13 and 14). The reciprocal of the mean inter-event interval ($1/\tau_{on}$) is proportional to the $Ni^{2+}$ concentration (Fig. 2e). However, the mean dwell time, $\tau_{off}$ demonstrates a negligible dependence on the $[Ni^{2+}]$. Similar measurements were also performed with $Co^{2+}$, in which the binding events appear as transient resistive pulses (Supplementary Figs. 15 and 16), reporting a mean dwell time $\tau_{off}$ of 1.32 ms, much shorter than was observed with $Ni^{2+}$, which was 220 ms (Supplementary Table 6). The rate constants $k_{on}$ and $k_{off}$ were respectively derived from results in Fig. 2e and Supplementary Fig. 15c. The equilibrium binding constant $K_b$ was calculated as:

$$K_b = k_{on}/k_{off} \qquad (3)$$

From the derived results, $Ni^{2+}$ demonstrates a significantly stronger binding affinity to a TAZ than $Co^{2+}$ (Supplementary Table 6). When $Ni^{2+}$ and $Co^{2+}$ were simultaneously examined, their binding events were unambiguously recognized (Supplementary Fig. 17). The PNRSS strand 14TAK has thus demonstrated its generality as a template to introduce any azides. When reacted with the 3-azidopropylamine, the generated TAZ itself can serve as a fixed reactant, to which $Ni^{2+}$ or $Co^{2+}$ will bind.

In most circumstances involving CuAAC, a TAZ is however treated as a linker, to which other functional modules can be attached[47]. Phenylboronic acid (PBA), the core component in

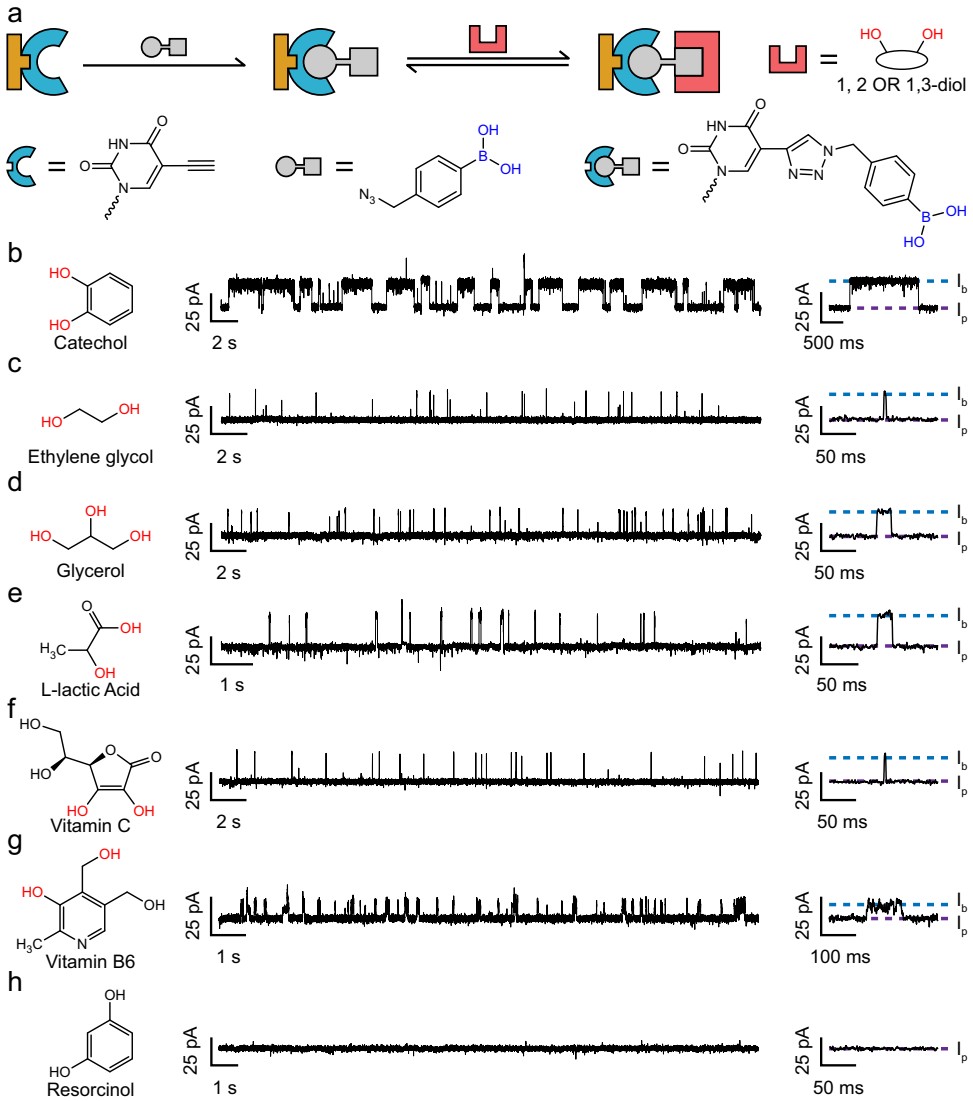

**Fig. 3 PNRSS with phenylboronic acid. a** Introduction of phenylboronic acid (PBA) and the corresponding reaction mechanism. A PBA was introduced with CuAAC as described in Supplementary Figs. 18–20. **b–g** The PBA on 14PBA serves as the fixed reactant. Mobile reactants containing 1,2-diols or 1,3-diols, such as catechol (**b**), ethylene glycol (**c**), glycerol (**d**), L-lactic acid (**e**), vitamin C (**f**), or vitamin B6 (**g**), all report clear and distinct binding events. **h** Resorcinol, which lacks a compatible reactive group, fails to report any binding events. The experiments were carried out as described in 'Methods'. Catechol (500 μM, **b**), ethylene glycol (18 mM, **c**), glycerol (12 mM, **d**), L-lactic acid (5 mM, **e**), vitamin C (1.6 mM, **f**), vitamin B6 (50 μM, **g**), or resorcinol (1 mM, **h**) were added to *trans* reaching the aforementioned final concentrations. Corresponding scatter plots of $\triangle I$ vs. $t_{off}$, results of $1/\tau_{on}$ and $1/\tau_{off}$ against different analyte concentrations and time histograms are provided in Supplementary Figs. 21–32.

sensors of saccharides or catecholamine[48], reacts with compounds containing a 1,2-*cis*-diol or a 1,3-*cis*-diol moiety to form a five-membered or six-membered boronate ester (Fig. 3a)[49–51]. To introduce a PBA to a PNRSS strand, 4-(azidomethyl)benzeneboronic acid was reacted with the PNRSS strand 14TAK by CuAAC ('Methods' and Supplementary Figs. 18 and 19). The product was further characterized by mass spectrometry (Supplementary Fig. 19) and single-channel recordings (Supplementary Fig. 20 and Supplementary Table 7), which confirmed the success of the conjugation and generated a new PNRSS strand referred to as 14PBA (Supplementary Table 1). To perform PNRSS, the PBA at site 14 of 14PBA serves as the fixed reactant. A +160 mV potential was continuously applied. Polyols such as catechol[50] (Fig. 3b and Supplementary Figs. 21 and 22), ethylene glycol[51] (Fig. 3c and Supplementary Figs. 23 and 24), glycerol[51] (Fig. 3d and Supplementary Figs. 25 and 26), L-lactic acid[50] (Fig. 3e and Supplementary Figs. 27 and 28), vitamin C[52] (Fig. 3f

and Supplementary Figs. 29 and 30) or vitamin B6[53] (Fig. 3g and Supplementary Figs. 31 and 32), serve as the mobile reactant and were individually added to *trans* at a desired concentration. Although the binding characteristics and kinetics differ (Supplementary Tables 8 and 9), all the aforementioned reactants reported detectable binding to a PBA. In contrast, when probed by 14TAK, no binding events were observed, confirming that the events registered the result of binding to the PBA (Supplementary Fig. 20e). Resorcinol, which is structurally incompatible with reaction with a PBA, failed to produce any binding events (Fig. 3h), further confirming the reaction mechanism. All observed reactions discussed in Fig. 3b–g report positive events ($I_b > I_p$). This is counterintuitive, because a bound molecule, which occupies more space in the nanopore lumen, is expected to produce negative events ($I_b < I_p$). A proposed mechanism is that binding of any analyte described above to a PBA results in the generation of an anionic boronic ester, as reported in several

previous literatures[54,55]. This generated negative charge may generally enhance the ionic flow through the pore. An evidence to approve that introduction of negative charges to the pore constriction would enhance the pore conductance is that the wild-type (WT) MspA, which has more negative charges on the pore constriction than that of M2 MspA, is significantly more conducting than M2 MspA in the open pore state[7]. On the other side, the overall size of the bound analyte may contribute to the reduction of the ionic flow. Thus, the overall contribution of a bound analyte to the blockage amplitude may be positive, especially when the studied analyte is a small molecule. A systematic study using quantum chemistry and molecular dynamics simulations may be carried out to further quantify this phenomenon but in a separate follow-up study.

A core advantage of nanopore-based single-molecule chemistry is that transient appearance of chemical intermediates can be probed, at a μs-resolution[56]. Tris(hydroxymethyl)-aminoethane (tris) is a widely used buffer reagent, whose conjugate acid has a pKa of ~8.3[57]. When dissolved in an aqueous solution at a pH near to its pKa, protonated or deprotonated forms of tris would both exist in significant proportions. In either form, a tris molecule, as a polyol, can react with a PBA and chemical intermediates resulting from its protonation or deprotonation may be observed. Although tris is easily accessible and widely used, tris binding to a PBA appears not to have been investigated to date. To demonstrate the direct observation of chemical intermediates with PNRSS, PBA and tris were applied as the fixed and the mobile reactant, respectively (Supplementary Figs. 33 and 34). At pH 7.0, binding of tris to PBA results in positive events, reaching the level $I_{b1}$. At pH 8.0, however, an extra binding level ($I_{b2}$) was observed, in addition to $I_{b1}$. Dynamic switching between $I_p$, $I_{b1}$ and $I_{b2}$ was also observed, where $I_{b1}$ and $I_{b2}$ represent the protonated and the deprotonated tris, respectively, when bound to a PBA (Supplementary Fig. 35). The observation that the deprotonated state, which is more negatively charged, reports a higher blockage state ($I_{b2} > I_{b1}$) is also consistent with our previous speculation that more negative charges generally enhances the ionic flow in this measurement configuration. Demonstrated with this simple example, direct observation of chemical intermediates enables an understanding of transient chemical processes. When properly designed, the appearance of chemical intermediates also helps to discriminate between subtly different chemical compounds. The above demonstration also suggests that a tris buffer should be avoided in any PBA-based polyol sensing assay by PNRSS so as to avoid interferences.

According to the previously reported nanopore-based single-molecule chemistry measurement[58], only a sole reactive site was permanently fixed in the pore lumen, meaning that any irreversible chemical reaction at this site would immediately terminate the production of any new information of chemical processes. Although rarely discussed, this technical restriction limits nanopore single-molecule chemistry studies to only reversible reactions. However, with PNRSS, the strand containing the fixed reactant is chemically separated from the pore. Even if it is irreversibly reacted, the whole PNRSS strand can be voltage rejected and reloaded, re-initiating a new cycle of measurement.

To demonstrate this, hydrogen peroxide ($H_2O_2$), a strong oxidant capable of irreversibly oxidizing a PBA to a phenol[59], was used as the mobile reactant (Fig. 4a) and a PBA was applied as the fixed reactant. During PNRSS, $H_2O_2$ initially reacts reversibly with the PBA, generating positive going spiky events, likely resulting from an intermediate state prior to the production of a phenol (Fig. 4a(ii)) as proposed in literatures[60,61]. However, the boron atom is not yet removed at this stage. Later, these spiky events suddenly disappear, giving a fluctuation-free baseline (Fig. 4b). In this state, this strand can still react with $Ni^{2+}$,

indicating that the TAZ linker is still present. However, it can no longer react with any polyol or $H_2O_2$, indicating that the boronic acid group has been lost, resulted from being irreversibly oxidized to a phenol (Supplementary Fig. 36). However, a new measurement cycle can be re-initiated by a voltage protocol (Fig. 4c) so that an irreversible single-molecule reaction can now be repetitively monitored, acknowledging this unique property of PNRSS (Fig. 4d and Supplementary Movie 3).

Besides the measurement of binding kinetics, rich information of chemical reactions monitored by PNRSS is also useful in recognition of single molecules. Norepinephrine, epinephrine and isoprenaline are catecholamine derivatives[62]. Norepinephrine and epinephrine are both natural hormones and neurotransmitters, and can be used medically[63]. Isoprenaline is a sympathomimetic β-adrenergic agonist medication[64]. Specifically, norepinephrine acts mostly on α-receptors and serves to maintain blood pressure, whereas epinephrine less specifically stimulates both α- and β-receptors and serves to relax the breathing tubes and to regulate blood flow, heart rate and glycogen metabolism. Isoprenaline, however, is used mainly in the treatment of bradycardia, heart block and asthma. These functional differences result from subtle variations in their chemical structures. However, they all contain a 1,2-benzenediol moiety, which reacts with a PBA (Fig. 5a).

PNRSS measurements of these compounds were carried out with 14PBA. Norepinephrine (Supplementary Figs. 37 and 38), epinephrine (Supplementary Figs. 39 and 40) or isoprenaline (Supplementary Figs. 41 and 42) were respectively added to *trans*, reaching the desired final concentrations. With a +160 mV potential continuously applied, all three compounds report negative proceeding binding events ($I_b < I_p$), dramatically different from that of the catechol that reports positive proceeding events ($I_b > I_p$). It is speculated that norepinephrine, epinephrine and isoprenaline, which are all cationic, may have compensated the effect of enhanced ionic flow due to the generation of the anionic boronic ester. This difference was more clearly demonstrated when catechol and norepinephrine were simultaneously sensed (Supplementary Fig. 43 and Supplementary Movie 4). Although their binding affinities are similar (Supplementary Table 10), binding events caused by norepinephrine, epinephrine or isoprenaline are however distinguishable by their distinct binding characteristics, including $\triangle I$, $\tau_{off}$ or noise levels. The acquired traces were first split by frequency into a low-pass and a high-pass portion, performed by a Butterworth filter with a 100 Hz cut-off frequency (Supplementary Fig. 44). The low-pass portion of events caused by norepinephrine binding reports $\triangle I$ of a smaller amplitude. The bound state is also free of any additional fluctuations. In contrast, events caused by epinephrine or isoprenaline binding result in $\triangle I$ of a larger amplitude, in addition to which secondary telegraphic fluctuations were observed (Fig. 5b). Events caused by epinephrine or isoprenaline can however be more clearly separated from their high-pass portion, in which binding of isoprenaline produces a high-frequency noise with a much larger amplitude. These differences in binding characteristics are more clearly demonstrated in the simultaneous sensing assay (Fig. 5c, d and Supplementary Movie 5). The events caused by binding of different catecholamines were efficiently distinguished by the SD value of the low-pass and the high-pass portion. For automatic event identification, these two parameters were applied to build a machine learning algorithm ('Methods' and Supplementary Fig. 45). Briefly, the algorithm was developed based on the support vector classification model, a machine learning method for data classification[65]. A total of 1455 events acquired with epinephrine, norepinephrine or isoprenaline as the sole analyte were applied to finalize the model. According to the confusion matrix results, events caused by isoprenaline were reported with an impressive 99.9% accuracy. Those caused by

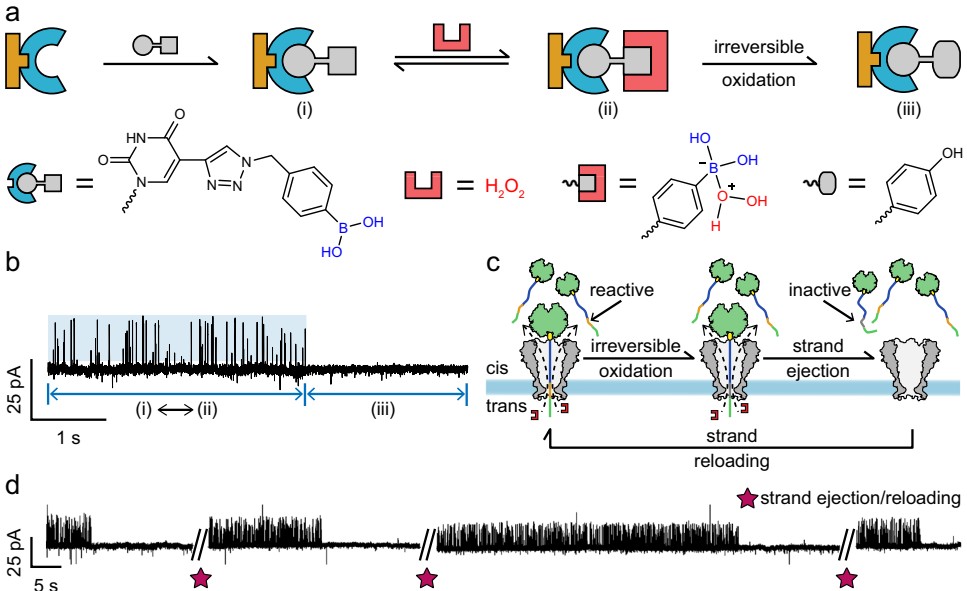

**Fig. 4 Repetitive PNRSS measurement of irreversible reactions. a** Introduction of PBA and the corresponding reaction mechanism. A PBA is introduced by CuAAC as described in Supplementary Figs. 18–20. The PBA on the PNRSS strand serves as the fixed reactant. Hydrogen peroxide, acting as a mobile reactant, may either reversibly bind to PBA or irreversibly oxidize the PBA to a phenol. **b** A trace containing the reversible (i–ii) and irreversible (iii) reactions of a PBA. **c** The PNRSS strategy to cope with irreversible reactions. Upon irreversible oxidation, the docked PNRSS strand is voltage ejected and reloaded to initiate a new measurement cycle. **d** A trace containing repetitive PNRSS measurement. Four consecutive cycles of measurements are demonstrated. The star label marks the moment when the voltage ejection and reloading was performed (Supplementary Movie 3).

epinephrine or norepinephrine were reported with a satisfactory 95.5% or 98.6% accuracy (Fig. 5e). When simultaneously sensed, events from a continuously recorded trace 15 min in length were extracted, and the low-pass and the high-pass SD values were presented (Fig. 5f). Three event populations, from binding of norepinephrine, epinephrine or isoprenaline, were clearly separated (Supplementary Fig. 46). A decision boundary plot generated by the machine learning algorithm was placed above the scatter plot to assist event recognition. The above demonstration with the three catecholamines has provided concrete evidence that the wealth of information generated by chemical reactions can facilitate single-molecule recognition. Although the reaction between a PBA and catecholamine is well understood[66,67], it is the first single-molecule chemistry study on the topic to be studied and was facilitated by PNRSS. Different from a previous report of neuron transmitter sensing by nanopore[68], in which only a weak amplitude of a 0.9 and a 1.1 pA were reported from epinephrine and norepinephrine, respectively, binding to an engineered α-HL, this approach with PNRSS applied a different chemical reaction and a conical pore, resulting in richer sensing information and a much larger event amplitude (~21–32 pA), clearly distinguishing three catecholamines (Supplementary Table 11). Frequency split analysis assisted by machine learning has further boosted the sensing performance. Distinct from the strategy of fluorescence probe design applied in ensembles, in which chemical synthesis of complicated probe structures are necessary to distinguish chemically similar catecholamines[69,70], the strategy by PNRSS requires only a sole PBA to distinguish three catecholamines. Other catecholamines, such as dopamine and L-3,4-dihydroxyphenylalanine, which are precursors of norepinephrine and epinephrine[71], were also studied by this system. However, these results will be reported separately in a follow-up study.

A variety of compounds based on nucleoside analogues have been synthesized, screened and clinically tested in the development of anti-viral drugs[72]. Remdesivir, a specific nucleoside analogue and an investigational anti-viral drug[73], has been reported to be effective in treating conditions caused by 2019 coronavirus (COVID-19), the agent responsible for the current pandemic that has caused a global crisis[74–76]. Remdesivir, a prodrug, is metabolically converted in cells into its active triphosphate form, which acts to block the RNA-dependent RNA polymerase, precluding the virus from further replication[77].

Remdesivir and its triphosphate metabolite both contain a ribose moiety, which reacts with PBA[78] (Fig. 6a). The corresponding PNRSS assay was designed with 14PBA placed in cis. Remdesivir (Supplementary Figs. 47 and 48) or remdesivir triphosphate (Supplementary Figs. 49 and 50) were treated separately as the mobile reactant and were respectively added to trans at the desired concentrations. A +160 mV potential was continuously applied. During PNRSS, both remdesivir and remdesivir triphosphate report positive proceeding events. However, events from remdesivir are long-resident and intensive fluctuations were also observed, whereas events of remdesivir triphosphate are much shorter resident and negligible level fluctuations were observed (Fig. 6b and Supplementary Tables 12 and 13). These distinct event characteristics are clearly demonstrated when the SD of the blockage level is plotted against the even dwell time (Fig. 6c). Distinction of the events is also demonstrated by the scatter plot of the high-pass and the low-pass SD values, from which two populations are separated unambiguously (Fig. 6d and Supplementary Fig. 51). A continuous trace performed with both reagents added is demonstrated in Fig. 6e, in which differences between both event types can be clearly recognized (Supplementary Movie 6).

Although there are conflicting conclusions on the therapeutic effect of remdesivir in treating COVID-19[79,80], the above demonstration has nevertheless expanded the types of analytes that can be investigated by PNRSS. Although the chemical reaction between PBA and nucleoside analogues has been previously investigated[78], binding between remdesivir and its derivative to a PBA has not been studied to date. With PNRSS, their single-

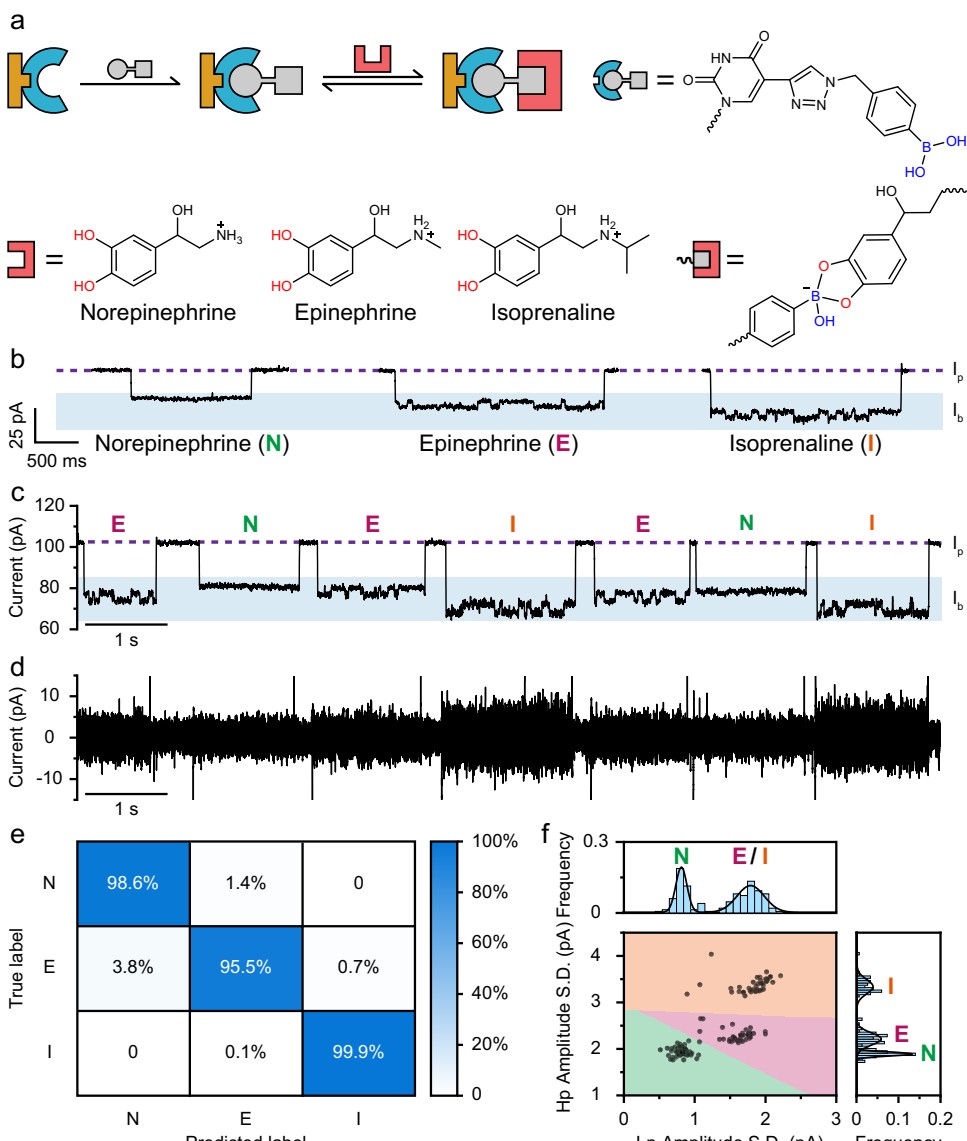

**Fig. 5 PNRSS sensing of epinephrine, norepinephrine and isoprenaline. a** The reaction mechanism. Norepinephrine, epinephrine and isoprenaline all contain a 1,2-benzenediol moiety, capable of binding a PBA. **b** Representative events of epinephrine, norepinephrine or isoprenaline binding. The events were low-pass Butterworth filtered with a cut-off frequency of 100 Hz (Supplementary Fig. 44). All events appear negative going ($I_b < I_p$). **c, d** A representative trace containing norepinephrine-, epinephrine- or isoprenaline-binding events. The trace was Butterworth filter separated into the low-pass (**c**) and the high-pass (**d**) portion. The cut-off frequency is 100 Hz (Supplementary Fig. 44). **e** A confusion matrix was generated based on 1455 events fed into a SVC model (Supplementary Fig. 45). **f** The scatter plot of low-pass (Lp) and high-pass (Hp) amplitude SD from PNRSS sensing of a mixture of catecholamine events (Supplementary Fig. 46). Results of 150 events are included. The decision boundary, which separates the scatter plot to green (norepinephrine), pink (epinephrine) and orange (isoprenaline) colour-coded regions, was determined by a machine learning algorithm (Supplementary Fig. 45).

molecule binding kinetics are determined, in a nano-confined space (Supplementary Table 13). Direct distinguishing of remdesvir and its metabolite is also achieved. Although not demonstrated in this study, other nucleoside analogues such as Galidesvir[81], Ribavirin[82] or Favipiravir-RTP[83] may in principle be recognized by a similar PNRSS assay. These demonstrations may inspire pharmacokinetics or drug-screening applications and may be useful in the current pandemic.

## Discussions

Twenty analytes have so far been analysed using MspA as the PNRSS pore. In principle, any nanopore in which a synthetic polymer could be tethered and fully stretched would be suitable to perform PNRSS. However, the performance is to be determined

by the overall structural of this pore. To approve the generality of PNRSS to work with other channel proteins, a feasibility test was performed using WT α-hemolysin (α-HL) as the PNRSS pore and 14PBA (Supplementary Table 1) as the PNRSS strand. Iso-prenaline was applied as the mobile reactant (Supplementary Fig. 52). Experimentally, events of isoprenaline binding, which appear as negative proceeding ($I_b < I_p$) resistive pulses, were successfully observed, confirming our hypothesis that PNRSS has a generality when used with other channel proteins. However, the reported event amplitude (~−4.6 pA) is much smaller than that produced by MspA (~−32.2 pA and Supplementary Table 11), although other measurement conditions were kept identical. This experimentally suggests that MspA, which has an overall conical geometry, results in a more focused electrical field at the pore constriction; thus, a larger event amplitude was produced. It is

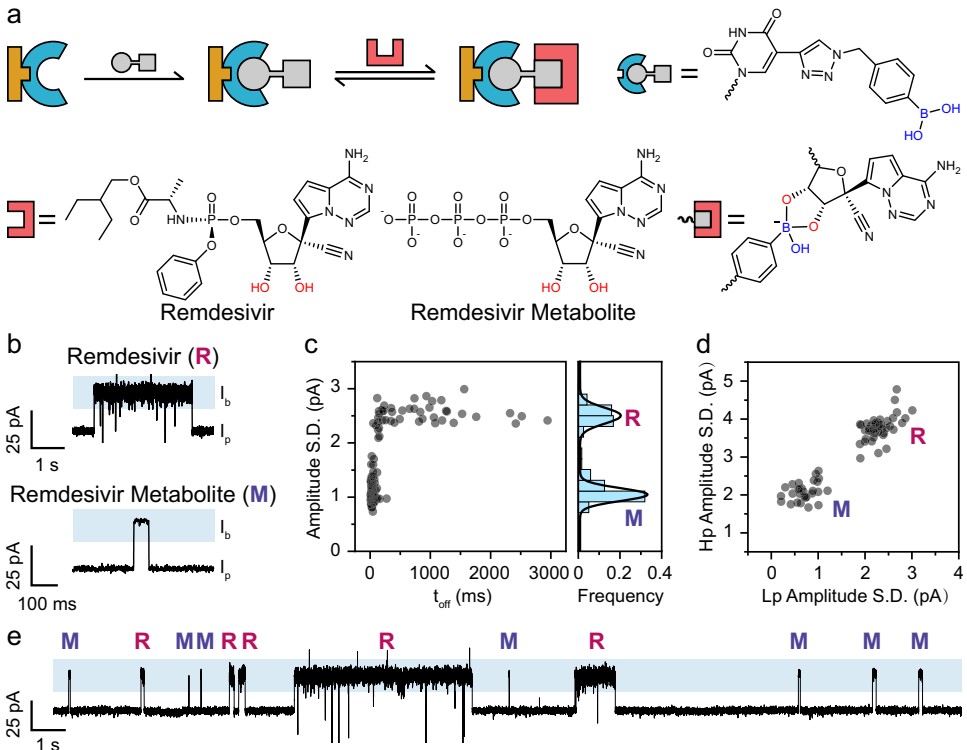

**Fig. 6 PNRSS sensing of remdesivir and remdesivir triphosphate metabolite. a** The sensing mechanism. Remdesivir and the remdesivir metabolite both contain a ribose moiety, capable of binding to a PBA. **b** Representative events of remdesivir and remdesivir metabolite. The label R and M respectively marks remdesivir and remdesivir metabolite. The events were Butterworth low-pass filtered with a cut-off frequency of 100 Hz (Supplementary Fig. 44). Both events appear positive going ($I_b > I_p$). Binding of remdesivir results in time extended events with characteristic noise fluctuations. Remdesivir metabolite on the other hand reports transient events with minimum noise. **c** A scatter plot of event amplitude SD against the event dwell time. The histogram of the event amplitude SD and its Gaussian fitting results were plotted to the right. Results of 119 events are included. **d** A scatter plot of high-pass (Hp) and low-pass (Lp) amplitude SD. Both analytes are clearly separated in the scatter plot. Results of 126 events are included. **e** A continuous trace containing binding events from remdesivir and remdesivir metabolite. The identity of each event is called based on event characteristics as described in **d**.

thus an optimum choice of PNRSS pore to distinguish between analytes with similar chemical structures. All components of the PNRSS strands are indispensable. A PNRSS strand with no traction section (14TAK-NTS, Supplementary Table 1) was as well tested, which fails to report any successful trapping of the PNRSS strand (Supplementary Fig. 53). It has thus confirmed the importance of the traction section to maintain an electrophoretic force during the measurement.

To be consistent all through the study, the measurement condition was generally kept identical with which a buffer of 1.5 M KCl, 10 mM HEPES was used and a high voltage such as +180 or +160 mV was applied. By respectively taking catechol (Supplementary Fig. 54) or norepinephrine (Supplementary Fig. 55) as representative electrical neutral or electrical positive analyte, PNRSS measurements were carried out with a gradient of voltages between +80 and +160 mV. For both analytes, the reported event amplitude is generally larger when a higher potential was applied, suggesting that a higher applied potential is more advantageous by producing a higher sensing resolution. Although a +20 mV potential is enough to maintain the PNRSS strand in the pore lumen and backflow of the PNRSS strand is rarely observed, a minimum of +60 mV potential was required to produce a large enough event amplitude to be detectable. The event dwell time is generally independent of the amplitude of the applied potential for both analytes. However, norepinephrine, which is positively charged, reports a clearly higher rate of event appearance when a larger potential was applied, indicating that the electrophoretic force is critical when the mobile reactant is charged (Supplementary Fig. 55). In contrast, the rate of event

appearance is less modulated by the applied potential for catechol, approving that the contribution of the electroosmotic flow is negligible (Supplementary Fig. 54). PNRSS may also be carried out at different salt concentrations and generally a higher salt concentration is also more advantageous to produce a large event amplitude by producing a larger flow of ions through the pore (Supplementary Fig. 56).

The rate of single-molecule chemical reactions may as well be modulated by temperature. Experimentally, by taking norepinephrine as a model analyte and PBA as the fixed reactant, PNRSS was carried out on an Orbit Mini nanopore reader (Nanion Technologies GmbH, Germany) with a built-in temperature control module (Supplementary Fig. 57). Clearly, both the on and off rate and the binding affinity is modulated by the temperature. The reaction rates are generally exponentially related with the set temperature, which fit the rule as described by an Arrhenius equation[84]. The *trans* compartment of the Orbit Mini chip is too small to place the mobile analyte; thus, in this measurement, norepinephrine was added to *cis*, different from that performed in Supplementary Fig. 37. However, events were detectable in both configurations, confirming that fast diffusion of the small-molecule analyte is also contributing to the generation of events.

In this study, the limit of detection is specifically defined as the minimum final analyte concentration in the measurement chamber to acquire at least five events during a 10 min continuous measurement (Supplementary Table 14). Due to the large volume of the measurement chamber and the small size of a single nanopore sensor, the efficiency of detection is generally not

optimum for the current configuration. Thus, without any sample enrichment, it is not expected to observe enough events for quantification when a target analyte was present in a low concentration in a physiological sample. This is the case for epinephrine, norepinephrine and isoprenaline, which have a ~ nM physiological concentration in the blood serum[85]; however, the detection limit reported here is ~1 μM. Engineering wise, this may be improved by introducing thousands of independent parallel nanopore sensors and an extremely flat flow cell to boost the sensing efficiency, similar to a configuration demonstrated in a MinION sequencer[19]. On the other side, some analyte tested in this study may appear at a high concentration in natural samples and may be directly applied for detection even with the current setup. Vitamin B6 reports an easily recognized event pattern during PNRSS and the detection limit is ~400 nM. We thus designed an assay to mimic detection of Vitamin B6 in true human urine samples[86]. Experimentally, addition of true human urine sample to *trans* did not report any interfering events (Supplementary Fig. 58). However, test of human urine samples with added vitamin B6 reported corresponding events, suitable for direct quantification according to the calibration curve reported (Supplementary Fig. 59). These results indicate that the PNRSS assay is ready to perform tests with true biological samples, similar to other state-of-the-art nanopore assays[87,88]. A unique event pattern is helpful to assist event recognition from a heterogeneous solution.

However, the above demonstration is still away from perfection. For quick demonstrations, all PNRSS were performed with the pore and the strand chemically separated. Although this sensing mode is advantageous for repetitive measurement of irreversible reactions, a PNRSS strand may be permanently conjugated to the pore, to further boost the resolution and consistency of PNRSS. The triazole produced by the CuAAC reaction may, e.g., report undesired chemical reactions with transient metal ions (Fig. 2). This interference may however be minimized by the addition of chelating agent such as ethylenediaminetetraacetic acid or by the choice of an alternative conjugation chemistry[89]. Results in this study might be inspiring in preparation of chemical compounds. However, due to the small scale of this single-molecule reactor, PNRSS is designed as a sensing instead of a preparative method at the moment. The $K_b$ values measured by PNRSS were also compared with those reported in literatures (Supplementary Table 15). It is however difficult to find results describing all chemical reactions reported in this study and performed at an exactly identical condition but a general consistency of result is seen. However, we would like to emphasize that the purpose to develop PNRSS is to apply existing knowledge of chemical interactions between reactants to achieve direct chemical sensing of small molecular analytes, which has been well supported by results in this study.

To summarize, PNRSS, first reported in this study, serves as a convenient molecular toolkit with which to study single-molecule chemistry processes with a nanopore. To better illustrate the idea of PNRSS, an artistic video demonstration (Supplementary Movie 7) was included. The specific aim is to break the technical bottleneck to introduction of any number or type of reactive groups into any spot of the nanopore lumen, which is difficult, even for the best in the field. With PNRSS, however, this difficulty has been transformed instead to synthesis of functional DNA oligomers, a routine performed daily by countless biochemistry labs or as a low-cost service provided by a variety of commercial vendors. Distinct from the previous configuration, PNRSS has also enabled repetitive monitoring of irreversible reactions, further broadening the choice of chemical reactions that, previously, were difficult to study by a nanopore. We report a sum of 20 single-molecule chemical reactions, in which hydrogen peroxide,

buffer reagent, transition metal ions, glycerol, lactic acid, vitamins, catecholamine derivatives or anti-viral medicines participate. The reported event patterns are highly diversified, associated with the size, charge and conformation of the analytes, useful for single-molecule recognition. Although limited by the length of this study, these reactions demonstrate core aspects of the PNRSS technique and show its feasibility and versatility. To the best of our knowledge, this is the largest number of nanopore-based single-molecule chemistry reactions that have been reported in a single publication. Most of them have never been previously investigated as single molecules, as summarized in Supplementary Table 16. Given the conical structure of MspA, which efficiently focuses ionic flows to its narrow pore restriction, all single-molecule chemical reactions discussed in this study have demonstrated a fully resolved event amplitude. For PBA, direct recognition of epinephrine, norepinephrine and isoprenaline also suggests its immediate biomedical applications. The single-molecule discrimination of remdesivir and its triphosphate metabolite by PNRSS may inspire pharmacokinetics measurements with a variety of nucleoside analogue medicines.

In subsequent studies, PNRSS may be carried out with multiple fixed reactants on the same strand, thus increasing the complexity of sensing. Although demonstrated with MspA and α-HL[6], other biological nanopores such as aerolysin[8] or CsgG[9] are, in principle, also compatible with PNRSS, as long as a tethered polymer containing the designed reactive site can be fully stretched in the pore lumen. Nanopores such as ClyA[10], FraC[11], PlyA/B[12] or phi29 Connector[14] with large openings may as well be applied to probe chemical reactions involving larger or more complex mobile reactants, such as polysaccharides or cyclopeptides. However, these proposed plans to apply PNRSS with large channel proteins have not been carried out yet but may be inspiring to other colleagues in the field.

## Methods

**Nanopore preparation**. The gene coding for the monomeric M2 MspA mutant (D93N/D91N/D90N/D118R/D134R/E139K) was synthesized and inserted in a pet-30a(+) vector. The M2 MspA was expressed with *Escherichia coli* BL21 (DE3) and purified using nickel affinity chromatography (GE Akta Pure, GE Healthcare)[90]. The purified M2 MspA spontaneously oligomerizes into an axis symmetric, octameric form, ready for all PNRSS measurements in this study. The octameric M2 MspA is the sole nanopore used in this work. For simplicity, it is referred to as MspA throughout the study, unless otherwise stated.

The gene coding for the monomeric WT α-HL was synthesized and inserted in a pet-30a(+) vector. Preparation of heptameric WT α-HL was carried out by *E. coli* BL21 (DE3) expression and purified using nickel affinity chromatography (GE Akta Pure, GE Healthcare)[90].

The plasmid DNAs coding for M2 MspA and WT α-HL have been shared with access code MC_0101191 and MC_0068416 in the molecular cloud plasmid repository (https://www.molecularcloud.org/s/shuo-huang, GenScript, New Jersey). Citation is requested when publishing with this plasmid.

**Nanopore measurements and data analysis**. Nanopore measurements were carried out in a custom measurement chamber. A self-assembled lipid bilayer is formed by 1,2-diphytanoyl-sn-glycero-3-phosphocholine, separating the chamber into the *cis* and the *trans* compartments. Each compartment is filled with 500 μL electrolyte buffer of 1.5 M KCl, 10 mM HEPES pH 7.0 or 8.0. All measurements with the PNRSS strands 14X, 13G/14G, 14A, 14G and 14TAZ (Supplementary Table 1) were conducted at pH 7.0. When tris was not used as the mobile reactant, all measurements with the PNRSS strand 14PBA (Supplementary Table 1) were accomplished at pH 8.0. For all measurements with tris, the pH was adjusted to either 7.0 or 8.0. A pair of Ag/AgCl electrodes was respectively placed in *cis* and *trans* sides of the chamber, in contact with the aqueous buffer on each side. The Ag/AgCl electrodes were electrically connected to a patch clamp amplifier to form a closed circuit. By convention, the electrode in the *cis* compartment is electrically grounded, while the opposing electrode is the working electrode.

To obtain the mean blockage level ($\bar{I}_p$), a static pore blockage measurement was performed[15]. Briefly, with a single MspA inserted, the PNRSS strand was added to *cis* with a 20 nM final concentration. A voltage protocol of +180 or +160 mV (0.9 s) and −100 mV (0.3 s) was applied repeatedly and $I_p$ was measured when the +180 or the +160 mV potential was applied. A minimum of 500 $I_p$ events were

collected during each experiment. The events were fit to a Gaussian distribution. $\bar{I}_p$ was derived from the central position of the fitting. Three independent measurements were performed to obtain the mean and the SD of $\bar{I}_p$. To perform PNRSS, the desired PNRSS strand was added to *cis* with a 10 nM final concentration. A positive potential was continuously applied. The PNRSS events were recognized as further pore blockage events, on top of the $I_p$ level (Supplementary Fig. 4).

All electrophysiology recordings were carried out with an Axopatch 200B patch clamp amplifier. The acquired traces were digitized by a Digidata 1550B analogue-to-digital converter (Molecular Devices, UK) with a 25 kHz sampling rate and low-pass filtered with a corner frequency of 1 kHz. Experimentally, MspAs were added to *cis* for spontaneous single-pore insertion. With a single pore inserted in the membrane, the electrolyte buffer in *cis* was exchanged to avoid further pore insertions. All PNRSS measurements were conducted at room temperature (rt) (21 ± 2 °C). Nanopore events were extracted by the single-channel search feature of Clampfit 10.7 (Molecular Devices, UK). Further analysis was carried out in Origin 2019. All colour-coded scatter plots were generated by ggplot2, an R package used for data visualization.

**Streptavidin DNA conjugation**. To form streptavidin-tethered DNA complexes, DNA oligomers with a 5′-biotin-TEG modification (Supplementary Table 1) were incubated with streptavidin, with an equal molar ratio at rt for 10 min. During a PNRSS measurement, the formed streptavidin-tethered DNA complex was added to *cis* at the desired final concentration.

**Theoretical calculations of optimized binding configurations**. All theoretical calculations were performed with the Gaussian 16 package suite[91]. Geometry optimizations were carried out using density functional theory with the M06 functional[92,93]. The 6-31+G(d) basis set was employed for C, H, O, N and P atoms, whereas the LANL2DZ basis set, together with the related effective core potentials[94], was used for Ni atoms. The relative energy ($\Delta E$) between the systems of low-spin ($E_{LS}$) and high-spin ($E_{HS}$) state was computed from:

$$\Delta E = E_{LS} - E_{HS} \qquad (4)$$

Possible binding modes of $Ni^{2+}$ with low-spin (LS) and high-spin (HS) states were investigated, as shown in Supplementary Fig. 3. Two modes of $Ni^{2+}$ cluster with four or five $H_2O$ molecules were calculated for the following study. The computational results of relative energy, $\Delta E$, between different spin states, are listed in Supplementary Table 3. The HS states of $(dGMP)_2$-Ni-4wt and $(dGMP)_2$-Ni-5wt were $-44.13$ and $-36.85$ kcal/mol lower than those of LS states, respectively, indicating that the HS states are energetically favourable. The geometry of HS states displayed a 6-coordination octahedral structure with the distance between Ni and N (O) atoms of 2.0–2.1 Å. However, the octahedral geometry showed the distortion to some extent in LS states. The O-H…O and O-H…N hydrogen-bonding interaction plays an important role in the binding with the $Ni^{2+}$ cluster.

The binding energy ($E_b$) was calculated to investigate the binding ability of different modes. The binding energy was obtained by calculating the energy difference between the total energy of the complex system ($E$) and the sum of individual energy of the $H_2O$ ($E_{wt}$), the $Ni^{2+}$ ion ($E_{Ni}$) and two deoxyguanosine monophosphates ($E_{dGMP/dGMP}$), respectively, and was calculated from:

$$E_b = E - xE_{wt}E_{Ni} - E_{dGMP/dGMP} \qquad (5)$$

where $x$ is the number of $H_2O$ molecules. A $Ni^{2+}$ ion can bind with four $H_2O$ and two guanine molecules, showing the binding energy with $-45.33$ kcal/mol. For $(dGMP)_2$-Ni-5wt, a $Ni^{2+}$ ion can bind one guanine and five $H_2O$ molecules with the $E_b$ of $-46.07$ kcal/mol, in which one $H_2O$ molecule can bind the guanine via the N(7) atom by hydrogen-bonding interaction.

**The introduction of functional azides to a PNRSS strand**. An alkyne-containing DNA strand 14TAK (Supplementary Table 1) was applied as a universal PNRSS strand template, to introduce any functional azides. Functional azides, such as 3-azidopropylamine (Supplementary Figs. 11 and 12) or 4-(azidomethyl) benzeneboronic acid (Supplementary Figs. 18–20), were chemically conjugated by a Huisgen CuAAC reaction.

**The procedure to produce 4-(azidomethyl) benzeneboronic acid**. Next, 4-(azidomethyl) benzeneboronic acid pinacol ester (158 mg, 0.6 mmol) and methylboronic acid (360 mg, 6 mmol) were added to a 10 mL reaction tube, and dissolved in acetone (2 mL). After further addition of 0.1 M NaOH (2 mL), the resulting solution was stirred at rt for 12 h. Afterwards, 4 mL dichloromethane was added to the solution. The resulting mixture was poured into a separatory funnel to remove the organic layer. The pH of the solution was adjusted to 7 by titration with 0.1 M HCl. Then, 4 mL dichloromethane was added in water layer to extract the target product. The organic layer was washed with $H_2O$ to remove residual salts. The organic phase was further dried with solid $Na_2SO_4$. The organic solvent was removed with a rotary evaporator to collect the 4-(azidomethyl) benzeneboronic acid as a white powder (74.2 mg, 68% yield)[95]. The product was further

characterized by $^1H$ NMR spectroscopy to confirm the success[96] (Supplementary Fig. 18).

**Reporting summary**. Further information on research design is available in the Nature Research Reporting Summary linked to this article.

## Data availability
The authors declare that the data supporting the findings of this study are available within the article and its Supplementary Information files, or from the corresponding authors upon reasonable request.

## Code availability
Machine learning was performed by DarwinML 2.0, a commercially available artificial intelligence software developed by Intelligence Qubic Technology Co. Ltd. Operation of the software is based on a graphic user interface (GUI) and no code is required to perform the analysis. Free trial of the DarwinML 2.0 can be requested from darwin.ai@iquibic.net.

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

## Acknowledgements

We acknowledge Professor Hagan Bayley (University of Oxford) for valuable suggestions during preparation of the manuscript. We acknowledge Professor Zijian Guo, Professor Shaolin Zhu, Professor Congqing Zhu, Professor Jie Li and Professor Ran Xie in Nanjing University, and Mrs. Yiou Ma, Professor Daoqiang Zhang and Xiaoyu Guan in Nanjing University of Aeronautics and Astronautics for inspiring discussions. This project was funded by National Natural Science Foundation of China (Grant Numbers 31972917, 91753108 and 21675083) and supported by the Fundamental Research Funds for the Central Universities (Grant Numbers 020514380257 and 020514380261), Programmes for high-level entrepreneurial and innovative talents introduction of Jiangsu Province (Individual and Group programs), Natural Science Foundation of Jiangsu Province (Grant Number BK20200009), Excellent Research Program of Nanjing University (Grant Number ZYJH004), Shanghai Municipal Science and Technology Major Project, State Key Laboratory of Analytical Chemistry for Life Science (Grant Number 5431ZZXM1902), Technology innovation fund program of Nanjing University.

## Author contributions

S.H. and W.D.J. conceived the project. W.D.J. and C.Z.H. performed the measurements. Y.M.G. and J.M. performed the molecular simulations. G.R.Q. designed the machine learning algorithms. Y.Q.W., Y.L., S.H.Y., J.C. and S.Y.Z. prepared the MspA nanopores. W.D.J. and X.Y.D. performed the measurements with α-HL. W.D.J. and L.Y.W. performed the measurements at different temperatures. P.K.Z. set up the instruments. S.H. and W.D.J. wrote the paper. Y.Q.W. and W.D.J. prepared the supplementary videos. S.H. and H.Y.C. supervised the project.

## Competing interests

S.H. and W.D.J. have filed patents describing the PNRSS technology and its applications thereof. G.R.Q. is the founder of Intelligence Qubic Technology Co. Ltd, a company engaged in the development of artificial intelligence software interfaces. The authors claim no other competing interest.
