## [Peer Review File · Nature Communications]

The previous round of reviews was completed at another journal

Author's Response to Reviewers

Reviewer #1 (Comments for the Author):

Reviewer comments:

It is a laudable goal to study the rapid formation or breaking of chemical bonds between single molecules in a nanoreactor, and it is likely to pay off in manufacturing. However, despite my enthusiasm for the effort, the manuscript entitled, "Programmable Nano-Reactors for Stochastic Sensing (PNRSS) by Wendong Jia et al., does not deliver on this promise. This manuscript does not offer the proper context for the work nor is it truly innovative.

Author Response:

We thank reviewer 1 for your comment. We would like to emphasize that the core innovation of PNRSS is to break the technical hurdle to perform nanopore single molecule chemistry by introducing the programmable component into the pore lumen. We so far haven't seen any reported work describing a similar scheme or is as advantageous and easy as our method. Reviewer 1 is right that the demonstrated idea may be paid off in manufacturing of chemical compounds. However, we again emphasize that the core topic of our work focuses on single molecule sensing of small molecules or intermediates during the reaction. Chemical manufacturing by a nanopore is a brilliant idea but it is not the main topic of this manuscript.

Reviewer comments:

Finally, the implementation developed by the authors that relies on an MspA proteinaceous pore in a lipid bi-layer membrane affords inadequate temporal resolution to deliver on the prospect for "rapid formation or breaking of chemical bonds" as advertised in the abstract. Therefore, I do NOT recommend publication in Nature Nanotechnology.

Author Response:

We thank reviewer 1 for your time commenting on this work. Each measurement apparatus has a limit in the temporal resolution. It is the need of the measurement which defines a required temporal resolution. The dwell time of events being measured in this work range from ms to seconds in the duration, which is definitely resolvable by a patch clamp amplifier. We didn't intend to observe sub-microsecond intermediates or fs transition states. However, useful information such as the appearance of intermediate states or the identity of a specific chemical compound can be obtained by

PNRSS. These are demonstrated by direct distinguishing between chemical compounds like neuron transmitters or COVID 19 medicine Remdesivir and its metabolite. The temporal resolution is clearly sufficient and the programmable nature of the PNRSS technique has been well demonstrated. I am confident that future engineering of this approach would push the temporal resolution of a patch clamp amplifier further, however it is not the main topic of the current study.

Besides, we would like to emphasize that the application of an MspA proteinaceous pore in a lipid bi-layer membrane has little to do with the temporal resolution of the measurement. It is more about the patch clamp amplifier which defines the temporal resolution. Thus, **all** nanopore single molecule sensing measurements applying an Axon 200B amplifier share the same maximum measurement bandwidth (100 kHz). However, very few reports have fully used this maximum bandwidth due to an unnecessary need or an undesired signal to noise ratio. Actually, measurement with an MspA proteinaceous pore in a lipid bi-layer membrane offers a low noise (I_{RMS}) and large event amplitude (ΔI) acknowledging its highly rigid structure and conical geometry, which enables measurement with a higher signal to noise ratio ($SNR = \Delta I / I_{RMS}$), more advantageous than those probed by solid state nanopores or aHL nanopores and may thus be more advantageous to gain a better temporal resolution. However, we still didn't find it necessary to further push the temporal resolution for the analyte being investigated in this work, considering that they have been fully discriminated from each other already.

Reviewer comments:

In this manuscript, Jia et al. outline a strategy to study of single molecule chemistry that uses a strand of a synthetic polymer with a streptavidin-tether site that is docked inside a nanopore. The chemical reaction is detected by monitoring the blockade current through the pore as a mobile reactant binds to a fixed reactant embedded in the synthetic polymer. Twenty single molecule chemical reactions were examined to support the idea that chemical reactions between single molecules can be analyzed this way.

Author Response:

We acknowledge reviewer 1 for the summary of our work. Besides the summary, we would also like to emphasize that the main innovation of PNRSS is that it is the first programmable nanopore sensor being developed to monitor single molecule reactions. A huge technical hurdle of protein engineering has been avoided, which is an advancement in the field. I am not sure whether reviewer 1 has noticed that or not. Prior to this work, no group other than Hagan Bayley's has demonstrated engineering of a truly heterooligomeric nanopore to monitor single molecule chemical reactions. With PNRSS, this technical hurdle has been much simplified. By ordering a custom synthetic strand of DNA, anyone can now perform nanopore single molecule chemistry measurement with ease.

Reviewer comments:

In a chemical reaction, the reaction velocity is usually at the focus of attention, especially for a manufacturing process. The key elements affecting the reaction velocity are: 1. the design of the physical reactor that transports reactants to, and products from the reaction site; 2. the intrinsic reaction rate; and 3. the design of catalysts that enhance the intrinsic rate. The long reaction times encountered in conventional reactors, and the high cost and poor selectivity of the catalyst have frustrated the efficacy of chemical reactions until now.

The design and fabrication of nanometer-scale reactors (that incorporate catalysts engineered on a molecular-scale to produce highly efficient, selective reactions) is very appealing because it enables dynamic control over molecular placement in space and time to yield highly efficient, nearly “perfect” reactions. The reactions can be “perfect” because diffusion occurs on a time scale that is fast compared to the reaction rate, which promotes high efficiency. Thus, nano-reactors have been the subject of intense scrutiny because of their relevance to manufacturing and because they effectively mimic biological processes.

The overall reaction velocity can be decomposed into its elementary steps: 1. reactants in a fluid are transported to the surface of the solid and/or catalyst; 2. reactants diffuse to the active sites; 3. at least one of the reactants chemisorbs onto the surface, and then reactants rearrange and react to form products; 4. products desorb and diffuse away from the catalyst. One of these steps is usually rate-limiting. A nano-reactor practically eliminates mass transport limitations by confining the reaction to sub-femtoliter volumes ($1 \text{ fL} = (1000 \text{ nm})^3$). Since the reaction is confined to a nanometer length scale, diffusion is exploited to efficiently deliver reagents on a microsecond time-scale to a catalytic surface, which maximizes the value of structures with intrinsically high surface-to-volume ratios. Indeed, improved reactivity, nearly 100% conversion efficiency and nearly perfect reactor utilization in sub-fL volumes has already been observed over a wide range of reactions, including heterogeneous catalysis, enzyme kinetics, protein binding and electrochemical activity. Typical optimized batch processes yield reaction rates of $\sim 10^{-5} \text{ mol s}^{-1} \text{ mg}^{-1}$ and require reaction times of 30 min. to several hours. In contrast, a nano-reactor can achieve effective reaction rates of $\sim 10^{-3} \text{ mol s}^{-1} \text{ cm}^{-2}$, assuming a flow rate of $\sim 1 \text{ nL/s}$ through each reactor and a very modest reactor density of 10^6 cm^{-2} . While the kinetics cannot strictly be compared due to configurational differences, nano-reactors allow for continuous processing, promising orders of magnitude higher production rates versus traditional reactor geometries. Thus, the combination of fast diffusive transport, enhanced surface reactions and the concentration of products within fL volumes makes nano-reactors ideal for efficient chemical synthesis and industrial scale-up.

With this background, my first complaint is that the authors have not placed their work in the proper context. They are ignoring a vast (and ancient) literature on the subject. For a few examples, see:

1. S. Petrosko, R. Johnson, H. White and C. Mirkin, "Nanoreactors: Small Spaces, Big Implications for Chemistry," *J. Am. Chem. Soc.* 138(24), 7443-7445 (2016).
2. P. B. Zetterlund and D.R. D'hooge, "The Nanoreactor Concept: Kinetic Features of Compartmentalization in Dispersed Phase Polymerization," *Macromolecules*, 52, 7963-7976 (2019).
3. A.T. Bell, "The impact of nanoscience on heterogeneous catalysis," *Science*, 299(5613), 1688-1691 (2003).
4. R. B. Schoch, L. F. Cheow and J. Han, "Electrical Detection of Fast Reaction Kinetics in Nanochannels with an Induced Flow," *Nano Lett.* 7, 3895-3900 (2007).
5. V. Hessel, H. Löwe, A. Müller, G. Kolb, *Chemical Micro Process Engineering - Processing and Plants*, Wiley-VCH, Weinheim (2005).
6. P.L. Mills, D.J. Quiram, J. F. Ryley, "Microreactor technology and process miniaturization for catalytic reactions—A perspective on recent developments and emerging technologies," *Chem. Eng. Science*, 62(24), 6992-7010 (2007).
7. M.N. Kashid and L. Kiwi-Minsker, "Microstructured Reactors for Multiphase Reactions: State of the Art," *Ind. Eng. Chem. Res.*, 48, 6465–6485 (2009).
8. H.M. Chen, R.-S. Liu, M.-Y. Lo, S.-C. Chang, L.-D. Tsai, Y.-M. Peng, J.-F. Lee, "Hollow Platinum Spheres with Nano-Channels: Synthesis and Enhanced Catalysis for Oxygen Reduction," *J. Phys. Chem. C* 112, 7522-7526 (2008).
9. Z. Wang, T.L. King, S.P. Branagan, P.W. Bohn, "Enzymatic activity of surface-immobilized horseradish peroxidase confined to micrometer- to nanometer-scale structures in nanocapillary array membranes," *Analyst* 134, 851-859 (2009).
10. L. Nyholm, "Electrochemical techniques for lab-on-a-chip applications," *Analyst* 130, 599-605 (2005).
11. M.A.G. Zevenbergen, P.S. Singh, E.D. Goluch, B.L. Wolfrum, S.G. Lemay, "Electrochemical Correlation Spectroscopy in Nanofluidic Cavities," *Analyt. Chem.* 81, 8203-8212 (2009).
12. B. Zhang, Y. Zhang, H.S. White, "Steady-State Voltammetric Response of the Nanopore Electrode" *Analyt. Chem.* 78, 477-483 (2005).
13. V. Polshettiwar, B. Baruwati, R.S. Varma, "Nanoparticle-supported and magnetically recoverable nickel catalyst: a robust and economic hydrogenation and transfer hydrogenation protocol," *Green Chemistry* 11, 127-131 (2009).
14. L-P. Wang, A. Titov, R. McGibbon, F. Liu, V.S. Pande and T.J. Martinez, "Discovering chemistry with an ab initio nanoreactor," *Nature Chem.* 6, 1044-1048 (2014).

Author Response:

We thank reviewer 1 for your time for providing such comprehensive description of other approaches to design nano-porous materials as nanoreactors to promote chemical reactions. However, we would like to emphasize that the main topic of this

study is single molecule sensing instead of manufacturing, which has been clearly described in the title of this work “Programmable Nano-Reactors for Stochastic **Sensing**”. All through the manuscript, we haven’t even used the word “**Manufacturing**” at all. We can understand that reviewer 1 is so interested in the development of perfect nano-porous materials or a nano-catalyst which may be paid off for manufacturing. We are sorry that this work has a different focus. Besides, we thank reviewer 1 for the 14 suggested references. However, again, they are technically not directly relevant to the topic of this study. To address this issue, we have briefly revised corresponding descriptions of this manuscript to further emphasize that the main topic of this work is single molecule sensing instead of chemical manufacturing. We are pasting the corresponding discussion here for your reference.

“Results in this manuscript might be inspiring in preparation of chemical compounds. However, due to the small scale of this single molecule reactor, PNRSS is designed as a sensing instead of a preparative method at the moment.”

Reviewer comments:

Second, this work is derivative of the pioneering work by Braha et al. [which was dutifully cited by the authors in Nature Biotech. 18, 1005 (2000)]. Using essentially the same scheme, Braha et al developed a nanopore sensor from mutant alpha-hemolysin (α -HL) and built binding sites into the lumen of the barrel of the pore so that analytes were registered by their partial blockade of a transmembrane ionic current.

Author Response:

We acknowledge reviewer 1 for this comment. It is also great that reviewer 1 has recognized that we have dutifully cited the mentioned reference. Though the mentioned reference is a pioneering work in this research direction, the newly developed PNRSS technique however has thoroughly lowered the technical hurdle to achieve nanopore single molecule chemistry measurements. Consequently, a large variety of analyte can now be studied with ease. Acknowledging the conical geometry of MspA, the produced event amplitude is also much larger than that reported in the mentioned reference. In summary, we believe that it is quite clear that PNRSS is a huge advancement in the field of nanopore single molecule chemistry. We have also provided a side by side comparison for your reference (**Figure RL1.1**). The mentioned reference is a quite early one in the field. However, even to date, no other report has been demonstrated to thoroughly solve the technical challenge of the need of protein engineering and an insufficient event amplitude, to the best of our knowledge.

Figure RL1.1. A comparison between results in the literature and PNRSS. It is quite clear that a much larger diversity of analyte has been successfully measured using PNRSS. The reported event amplitude is also generally much larger than that performed by an α -HL mutant (2 pA vs 30 pA).

Reviewer comments:

Since Bayley's early report, the research has developed considerably. For example, Lee and Bayley have introduced unnatural amino acids to provide a large variety of reactive side chains into the lumen of a semi-synthetic proteinaceous nanopore to investigate a broader spectrum of single molecule chemistry [see: J. Lee and H. Bayley, "Semisynthetic protein nanoreactor for single-molecule chemistry," PNAS 112(45), 13768 (2015)].

Author Response:

We acknowledge reviewer 1 for this comment and another mentioned reference. In this reference, a native chemical ligation strategy was applied to prepare protein nanopore. To achieve that, a monomeric α HL was prepared as three truncated segments. The peptide segment was completely synthesized by peptide synthesizer whereas the other two segments were *E. Coli.* expressed. With this approach, all three

segments have to be successfully prepared. Afterwards, all segments have to be chemically conjugated and pray that the product can still fold properly. Afterwards, this monomer has to be assembled with other 6 unaltered subunits to form a heteroheptameric aHL. The gained advantage is that an alkyne was introduced to the pore lumen. However, the whole process is **EXTREMELY COMPLICATED and DIFFICULT**. It could represent the state-of-the-art of nanopore preparation but it is far from an easy technique that other colleagues are willing to use. Actually, I, as a postdoc in Bayley lab, had witnessed the finishing of this work. After leaving of Joongoo Lee, the first author of this PNAS paper, this project has been discontinued simply because it is too difficult to perform. I think the purpose to publish that work by Hagan is to demonstrate that native chemical ligation could be applied to make biological nanopore. It is however not a suitable strategy for most researchers but PNRSS is.

Whereas, on our side, introduction of an alkyne is extremely easy. Simply ordering a piece of synthetic nucleic acid and insert it into the pore is enough. We would also like to emphasize that synthetic nucleic acid can be obtained from a large variety of commercial vendors (IDT DNA, ATD Bio, Sangon, Genscript to list a few) and a much larger variety of corresponding research can be immediately carried out afterwards. The cost to obtain a batch of synthetic DNA sample is ~\$100. **However, it is hard to imagine that a company would be willing to make protein nanopores by native chemical ligation with a ~\$100 cost. Very likely, a company would deny this request simply because it is too difficult to perform.** We hope the above explanation should have convinced reviewer 1 the difference between the mentioned reference and our work. However, I believe our technique is much easier to be mastered by anyone in the field, which is a great technical advancement and contribution for the field.

The two methods were compared and demonstrated in **Figure RL 1.2**. It is clear that six steps of preparation are needed to prepare a semi-synthetic pore to monitor single molecule chemical reaction and the prepared yield is extremely low according to the gel results. However, on our side, it simply requires a single PNRSS strand, custom synthesized by a company.

We thank reviewer 1 for mentioning this reference. It describes a DNA assembly which is an interesting technique. However, the mentioned reference is not a relevant work in the field of nanopore single molecule sensing. **To deny the innovation of our work, this reference should contain at least one result of single channel recording (Figure RL 1.3).** It is hard for us to be convinced that a technique is better than ours without demonstrating any nanopore single channel recording results.

Fig. 1

Fig. 3

Fig. 2

Fig. 4

No single channel recording is performed.

Figure RL 1.3. A demonstration of core materials of the cited reference by reviewer 1. We argue that this reference didn't even include any single channel recording results. It actually belongs to research of a completely different field. It is far from evident to conclude that this work is a better work than ours without any single channel recording results.

Reviewer comments:

Moreover, the idea of using a tethered molecule in the pore is also not new. Detecting structural DNA variants with nanopore has been accomplished by using DNA molecules tethered to the membrane of MinION flowcell, for example [A. Norris, Cancer Biol. Ther., 17, 246–253(2016)].

Author Response:

We acknowledge reviewer 1 for your comments. However, we would like to emphasize that the main concept of this work is to probe single molecule chemical reactions or small molecule analytes. **We didn't even acclaim that our work is to detect structural DNA variants.** Thus, the mentioned reference is technically irrelevant to the main topic of this study and should not be used to deny the core innovation of our work. **It is not suitable to use a nanopore sequencing work to deny the innovation**

of a nanopore single molecule chemistry work.

Reviewer comments:

Likewise, an OmpG nanopore has been used to detect biotin-binding proteins by tuning the selectivity and sensitivity through adjustments of the ligand tether length. [M.A. Fahie, ACS Sens., 1, 5, 614–622 (2016).] The manuscript by Jai et al is just an elaboration on these themes.

Author Response:

Figure RL1.4. A comparison between results in the literature and PNRSS. With a different pore, a different configuration and a different set of analyte, we believe it is quite concretely clear that they are two different work with different focuses. The size of analyte applied in our study is at least an order of magnitude smaller than the mentioned reference. It is hard to imagine that these two works are following the same scheme or are comparable.

Thanks reviewer 1 for your comment. **We would like to emphasize that the mentioned reference used a different pore, a different scheme and targets a different set of analyte.** Nanopore single molecule chemistry mainly focuses on study of small molecule analyte or single ions. However, this work focuses on study of protein with biotin affinity. To demonstrate this difference, a reference figure is demonstrated (**Figure RL 1.4**). It is a great work but it is irrelevant to the main topic of our study. **Again, it is hard to be convincing to deny the innovation of PNRSS however using a reference that is irrelevant to single molecule sensing of small molecules.**

Reviewer comments:

Third, the scheme implemented by Jai et al. has inadequate temporal resolution for

actually resolving the chemical reactions (i.e. "the rapid formation and breaking of chemical bonds" described in the abstract) and scrutinizing intermediates. The response (rise- or fall-) time of the measurement apparatus was not mentioned in the manuscript. However, the METHODS section indicates that, "All electrophysiology recordings were carried out with an Axopatch 200B patch clamp amplifier. The acquired traces were digitized by a Digidata 1550B analog-to-digital converter with a 25 kHz sampling rate and low-pass filtered with a corner frequency of 1 kHz." The Axopatch 200B has a maximum bandwidth of 100 kHz and the Digidata 1550B can sample at a maximum rate of 500 kHz. The rise(fall)-time (τ) of the signal follows from the bandwidth (Δf) according to: $\tau = 0.35/\Delta f$. Thus, under optimal conditions (for high-speed detection), the best temporal resolution that the Axopatch 200B amplifier could offer would be about 3.5 μ s. However, with the bandwidth limited to 1 kHz by a filter (to improve the signal-to-noise ratio presumably), the temporal resolution degrades to 350 μ s. Chemical reactions occur on a large range of time scales, but the elementary steps involving valence electron dynamics and single-bond rearrangements usually take place on femtosecond to picosecond time-scales. On the other hand, some bimolecular processes, including redox reactions in liquid solution, occur on longer time scales of microseconds to milliseconds. The data in the supplement to this manuscript demonstrates only about 1-2 millisecond resolution, which is just not good enough. Other work has already shown similar resolution [see: G. Olivo et al., "Following a Chemical Reaction on the Millisecond Time Scale by Simultaneous X ray and UV/Vis Spectroscopy," *J. Phys. Chem. Lett.* 8, 2958 (2017).]

Author Response:

First of all, we acknowledge reviewer 1 for your comment on the temporal resolution of the technique. Reviewer 1 has also mentioned that "Chemical reactions occur on a large range of time scales". Discrete states of chemical reaction being observed in this paper are all above ms a time scale some can reach a few seconds, which is well observable by our setting of PNRSS. **It is hard to imagine that a 350 μ s temporal resolution, as derived by the reviewer himself/herself, can't resolve events lasting for a few ms to a few seconds.** We would also like to emphasize that nanopore techniques offer a better temporal resolution than other tools that can study single molecule chemistry, such as force spectroscopy (*Nat. Chem.* 11, 310–319 (2019)), single molecule Raman spectroscopy (*Chem. Rev.* 2017, 117, 11, 7583–7613), et al.

We understand that there are some quick chemical reactions that reviewer 1 is extremely interested to observe. However, it is not directly relevant to the demonstration of the programmable nature of our nanopore sensor but may be carried out in future study with a special emphasize on monitoring fast reactive states using nanopore.

Reviewer comments:

Moreover, the tether and/or the confinement associated with the pore could sterically interfere with the chemical reaction. For example, Zalineeva et al. found that electro-oxidation within nanopores (30–100 nm in diameter) prepared in Pd–Bi substrates differed markedly from that observed in bulk catalysts [A. Zalineeva, et al. J. Am. Chem. Soc., 136, 3937–3945 (2014)]. In the context of glycerol oxidation in alkaline media, the diffusion of products, intermediates, and reactants into and out of the nanopores was significantly perturbed; the pores permitted rodust selectivity as a function of potential. Likewise, Xiao and co-workers observed that confinement within a carbon nanotube shifts the optimal catalytic behavior to metals with a higher binding energy because the confined environment weakens substrate adsorption interactions [J. Xiao, et al., J. Am. Chem. Soc. 137, 477–482 (2015)].

Author Response:

We thank reviewer 1 for this comment. The tether is a few nanometers away from the reactive site, which in principle won't interfere the chemical reaction. It is possible that the narrow pore constriction may however interfere the measured kinetic constant so that a different number is reported in bulk. However, it is also this nanopore confinement which reports the single molecule data. Though some transient chemical intermediate may as well be observed by nanopore single molecule chemistry, the main purpose of this paper is not to measure the kinetic constants more accurately. Just the opposite, based on existing knowledge of chemical reactions, the introduced single molecule reactive site selectively and specifically recognizes different target chemical compounds and structure differences with an atomic resolution can as well be reported. We thank reviewer 1 for the comment and corresponding discussions have been added in the revised manuscript.

Reviewer comments:

Finally, one way to improve the report would be to provide a comparison between molecular dynamics simulations, including quantum mechanical features associated with binding, and high-speed measurements of a chemical reaction. Such work would be truly innovative.

Author Response:

Thanks reviewer 1 for the comments. Quantum chemistry simulation has been performed to describe Ni²⁺ binding to a dual guanine reactive site in our original submission (**fig S3**). We placed this simulation result to demonstrate how theoretical simulation could help us to understand the configuration of analyte when trapped in the nanopore constriction. However, still, the main topic of this study is to experimentally demonstrate the programmable nature of this nanopore sensor. Other thorough investigations using simulation would be carried out after this work. We still

acknowledge reviewer 1 for this suggestion.

Reviewer comments:

In summary, the manuscript entitled, “Programmable Nano-Reactors for Stochastic Sensing (PNRSS) by Wendong Jia et al., represents only an evolutionary elaboration of previous work. It does not offer the proper context nor is it truly innovative work. The implementation in an MspA proteinaceous pore affords inadequate temporal resolution to deliver to scrutinize chemical reactions. Because of these objections, I do NOT recommend publication in Nature Nanotechnology.

Author Response:

We acknowledge reviewer 1 for the summary. We have demonstrated an unreported strategy which has truly reduced the technical barrier to prepare a truly functional nanopore sensor which can resolve discrete states of chemical reactions. Acknowledging this advantage, a large variety of analyte has been studied, which successfully demonstrated the versatility of the technique. **To deny the core innovation of this technique, reviewer 1 only needs to find one suitable and previously reported work that fits the following criteria:**

1. A nanopore sensor that is **truly and fully programmable** in its reactive composition so that **ANY** chemical reactive sites can be introduced to **ANYWHERE** within the pore lumen. We would like to emphasize that any protein engineering-based strategy doesn't merit this need since every site-directed mutagenesis to a biological nanopore would experience a high risk that this protein may never be successfully prepared at all. Failed protein mutants are actually quite commonly seen. However, these failed examples can never be reported in a high impact paper, which may give the readers or reviewers an impression that protein engineering is free of any trouble. For example, mutagenesis to site 90 of MspA will damage protein assembly. This is however never a problem for PNRSS, which has gained the technique a great versatility and generality.
2. A report of such a strategy that doesn't require a large amount of **labor and cost investment**. Our strategy uses standard routines of synthetic ssDNAs which could be synthesized within days and could be provided by a large number of commercial vendors (Genescript, Sangon, IDTDNA, ATDbio, to name a few of them) all over the world. A batch of PNRSS strand costs <\$100 and could be ordered by any researcher without any need of previous training. One batch of PNRSS strand could also be used for ages. **It is hard to imagine that a custom engineered protein nanopore could be commercially provided with such a low cost.** Actually, most companies will deny the request of custom membrane protein synthesis simply due to its unpredictable nature. However, nucleic acid synthesis is never a problem and they are extremely conveniently accessible and cheap.

3. A report of a nanopore single molecule chemistry work with an equivalent versatility or a much better performance. We have demonstrated 5 PNRSS strands and 20 analytes within a single paper. This could not be easily achieved by protein engineering to nanopores.

Prior to this study, we believe that there is no previously reported nanopore reactor that can demonstrate a similar performance, especially the versatility and the event amplitude. Though reviewer 1 has listed some references, we have clearly explained why they are actually either not relevant to this study or is not advantageous than PNRSS.

Reviewer 1 may find some previous nanopore single molecule chemistry work. However, these works didn't even solve the problem to engineer the heterooligomeric porin. You will see many multiple step events corresponding to simultaneous happening of reactions on multiple sites. This is not desired and is not advantageous than PNRSS. To avoid delay of publication and for your references, we are listing these representative references for your reference (**Figure RL1.5**).

Figure RL 1.5 A demonstration of single molecule chemistry measurements acquired from homo-oligomeric nanopore sensors. Multiple steps of events corresponding of simultaneous reactions from multiple reactants were observed. We would like to emphasize that these demonstrations are far from more advantageous than PNRSS considering that the issue of generating a sole reactive site has not been successfully solved.

Eventually, we thank reviewer 1 again for providing so many insightful and inspiring information on this manuscript. The suggestion to monitor extremely fast chemical reaction states is a fantastic suggestion but more suitable as a future study. It is unfortunate that reviewer 1 didn't recognize the core innovation of this work. However, with our explanation in the response letter, we hope reviewer 1 now understand its innovation.

Figure RL1.6. An artistic demonstration of PNRSS. PNRSS is a highly modular technique. It is like a tool box, by combining different reactive modules, a large variety of chemical sensing function could be enabled. The more difficult protein engineering has been omitted. Please also don't miss watching Video S7 for an animation describing the PNRSS method.

Nanopore single molecule chemistry is a truly powerful tool, however not yet mastered by most in the field. To achieve that, we have developed PNRSS. PNRSS is like a highly versatile tool box containing useful modules, which could be combined to generate countless functions of nanopore sensing. However, troublesome protein engineering to the pore is not required. We sincerely demonstrate the concept of PNRSS using an artistic figure for your reference as well (**Figure RL 1.6**). **To better assist all readers to replicate this technique, the plasmid DNA coding for M2 MspA is also openly shared and can be obtained by anyone.**

Access link: <https://www.molecularcloud.org/s/shuo-huang>

Access ID: MC_0101191

We hope our humble effort could contribute the field this powerful tool and your support is highly appreciated.

Besides, we would also like to emphasize that core advancement of technologies such as DNA origami and CRISPR-cas9 has also used the programmable nature of nucleic acid to simplify tasks like fabrication of a large variety of nanostructures or gene editing (**Figure RL 1.7**). To make a nano scaled geometry or to edit a gene can be previously done by other methods. It is however the reduction of the technical hurdle by these methods which made the corresponding field thrived.

Figure RL1.7. A demonstration of other technologies that have applied the programmability of nucleic acid to gain highly versatile functions. It is hard to deny these innovations when engineered proteins instead of DNA nanostructures may as well be applied as nanomaterials. Similarly, ZFN and TALEN which are proteins that can edit genomes. However, it is CRISPR-CAS9 which has much lowered the technical hurdle of gene editing by incorporating a much more programmable RNA as its core component.

Reviewer #2 (Comments for the Author):

Reviewer comments:

Authors of this manuscript present a versatile strategy, “programmable nano-reactors for stochastic sensing” to democratize nanopore based single molecule chemistry investigations. Reading this manuscript conveys the impression of reading a patent or an experimental protocol. This manuscript entirely consists of vague and imprecise statements rather than explanation.

Author Response:

We acknowledge reviewer 2 for recognizing the versatile nature of this newly developed technology. To demonstrate its versatility, numerous examples were included, covering core aspects of how PNRSS may be applied in different sensing scenarios. The development of PNRSS has applied an extremely feasible strategy and a large variety of sensing can thus be performed. We aim to present as many details as possible so that readers interested in this technology can immediately replicate methods described in this paper. For this purpose, the plasmid DNA coding for the core protein nanopore in this paper is as well shared. I assume that a larger amount of experimental data and a rather detailed description of experimental methods may have given reviewer 2 the impression that the manuscript reads like a patent or protocol. However, we believe it is generally a good thing for any researcher to present more data and detailed experimental procedures to ensure the repeatability of their newly developed methodologies. Writing of this manuscript has also strictly followed the general criteria of a scientific paper and it has been polished by professional proof-reading services prior to submission. It is unfortunate that reviewer 2 didn't raise any more informative comments or suggestions. Still, based on comments from other reviewers, we have correspondingly revised the manuscript.

Reviewer #3

Reviewer comments:

The manuscript “Programmable Nano-Reactors for Stochastic Sensing” by Wendong Jia et al, studies several chemical reactions, for up to 20 single molecules, using a protein nanopore. To answer this challenge, the authors designed and manufactured an elegant nanoreactor with a MspA nanopore used to dock a streptavidin tethered PNRSS strand. A part of this strand contains a versatile reaction section with different kinds of reactive sites to perform the chemical reactions. This work's central claim is shown: the ability to use the nanoreactor as a versatile platform to measure the association/dissociation rates and binding constants of different chemical reactions at the single-molecule level. Another interesting claim is the ability to detect chemical intermediates.

Author Response:

We sincerely acknowledge reviewer 3 for recognizing the core innovation of PNRSS, which is to free the troublesome protein engineering effort to build an elegant nanoreactor for stochastic sensing. Besides the recognized claims listed by reviewer 3, we would also like to emphasize that the establishment of a reactive nanopore sensor also has a direct application on selective sensing of small molecules. The introduced reactivity along with a high sensing resolution of MspA has enabled the PNRSS system to directly discriminate between chemically similar compounds with no ambiguity. Direct sensing of vitamins, epinephrine, norepinephrine, isoprenaline, Remdesivir and its metabolite were great examples of this kind, as demonstrated in **Figure 3, 5 and 6** in the manuscript. As suggested by another reviewer (reviewer 4), direct detection of vitamin B6 in human urine samples was demonstrated in the revised manuscript as well, indicating that PNRSS may be properly engineered to suit the need of real clinical diagnosis. Acknowledging its programmable nature, future development of this technique will enable many more applications of this kind however corresponding effort of engineering has been much simplified.

Reviewer comments:

The authors claim that these programmable nanoreactors for stochastic sensing are in principle compatible with other protein channels but also for nanopores fabricated with solid state materials. To check that this approach is not limited to a single channel but a new general concept to study single-molecule chemical processes using nanopores, I strongly recommend performing an additional experiment with one or two species using another channel.

Author Response:

We sincerely acknowledge reviewer 3 for this comment. In our original manuscript, we suggested a general compatibility of PNRSS with other channels simply because success of DNA translocation has been intensively demonstrated by various nanopore sensors. If a piece of ssDNA could be fully stretched and the reactive site on the strand

can overlap with the pore constriction, PNRSS sensing should be equivalently carried out with other channel proteins. However, the performance of sensing using different pores may vary largely due to variance of the pore geometries.

Following the suggestion of reviewer 3 to perform PNRSS using another channel with at least one or two species, we have designed a corresponding experiment using wild type alpha hemolysin (WT α HL) and the PNRSS strand 14PBA. Isoprenaline was applied as the analyte. The demonstrated results have clearly approved that PNRSS can be carried out in WT α HL as well. However, the produced event amplitude is smaller than that produced by MspA, largely due to the non-advantageous cylindrical lumen structure of WT α HL. From this aspect, this result also serves as a proof that the conical structure of MspA is advantageous to be applied as the core of the nanoreactor.

The corresponding results were summarized in **fig. S52** in the revised manuscript accompanied with relevant discussions. This issue is thus well addressed. We are extremely grateful for this wonderful suggestion from reviewer 3 which has thoroughly approved the generality of PNRSS and the structural advantage of MspA. For your convenience, we are pasting the corresponding results here for your reference as well (**Figure RL 3.1**).

Figure RL 3.1. Brief demonstration of fig. S52. PNRSS measurements were carried out with WT α -HL as suggested by reviewer 3. It is expected to observe PNRSS events. However, due to the cylindrical instead of the conical geometry of α -HL, the reported event amplitude is smaller than that measured by MspA.

Reviewer comments:

The experiments are well performed, but occasionally, the number of events is insufficient to statistically validate the hypothesis, especially where only a few events were observed for the reaction intermediates.

Author Response:

We are extremely grateful for reviewer 3 for the comment that the experiments demonstrated in this paper were well performed. Being the PI of a junior research group, we truly treasure each opportunity of submission. All demonstrated measurements were seriously designed and performed. We apologize that some results may appear to have insufficient number of events. To address this issue, in the revised manuscript, all core results were accompanied with **SI tables** listing the number of events to draw statistical conclusions. We have also clearly noted the number of events for every scatter plot in the corresponding figure legends. Some measurements were carried out again and a much time extended trace was applied for data analysis, with which sufficient number of events were simultaneously demonstrated (**fig. S46**).

Reviewer comments:

The interpretation of some results must be more deeply discussed. For a broad audience, the introduction should provide a global overview of the state-of-the-art, including of all the challenging experimental systems that could be studied with nanoreactors. We note that although the state-of-the-art is partially introduced along with previous literature, significant references are missing. For publication in a journal of this standing, the manuscript needs numerous improvements,

Author Response:

We highly appreciate reviewer 3 for recognizing that the state-of-the-art has been partially introduced and we sincerely apologize for missing any significant references. Following a more detailed suggestion from reviewer 3, this part has been thoroughly improved in the revised manuscript. We sincerely thank reviewer again for this suggestion to improve the manuscript to fit a broader audience. We however hope reviewer 3 can understand that 82 references have been included in the revised manuscript already. We are restricted to include only those critically related to the topic of this study.

Reviewer comments:

as well as additional experiments to validate this nanoreactor concept using another nanopore.

I recommend rejecting the manuscript but resubmitting, if the suggested experiments are performed and all the issues raised are addressed.

Author Response:

We sincerely acknowledge reviewer 3 for an overall evaluation of this work and we truly treasure the given opportunity of revision. The suggested validation experiment using another nanopore has also been well carried out with WT aHL (**fig. S52**). Other issues were also fully addressed and have been summarized in a point by point report

below for your reference.

Reviewer comments:

Introduction and state of the art

The first part of the introduction should be improved with a more general description of the state-of-the-art, including the actual challenges and explaining how this work will address these challenges.

For instance:

“We here present ... to democratize nanopore based single molecule chemistry investigations.” This part is too vague. Is this approach only limited to the field of chemistry then? If not, what could other fields may also be studied with this type of system?

Author Response:

Thanks reviewer 3 for the valuable suggestion. The introduction part has been carefully revised to fit the request. The 2nd and 3rd paragraph of the introduction section should have now clearly explained the current societal challenges and why PNRSS is a suitable technique to address these challenges. We acknowledge reviewer 3 again for such valuable suggestion to improve this part of writing.

Generally, PNRSS is suitable to study small molecular chemical compounds and its related reactions in a single molecule scale. Besides chemistry study, many applications derived from small molecule sensing such as pharmacokinetics, drug screening, body liquid analysis, metal ion sensing and many others. With demonstrations in all core figures, we believe the feasibility of these acclaimed applications are well approved in corresponding demonstrations such as metal ion sensing, neuron transmitter sensing, drug metabolite sensing and vitamin B sensing in human urine samples.

Reviewer comments:

“However, investigations by α -HL generally report a weak event amplitude, measuring 1-5 pA”. The nanopore field is not limited to alpha-hemolysin. Other channels give better resolution! This part should be clarified. - I recommend adding a few lines on current societal challenges and how nanopores may be applied to solve these (cite a few papers).

Author Response:

We acknowledge reviewer 3 for this comment. Our original aim was to compare our work with previous single molecule chemistry studies and α -HL has been intensively studied on this purpose before. We apology for not including discussions of other great pores. Actually, with more than three decades of development, more and more types

of nanopores have emerged, forming a rather big and untied family. In the beginning of the 2nd paragraph, we have mentioned major biological pores in the field. We have also specifically explained that the 1-5 pA event amplitude results from only engineered aHL for single molecule chemistry measurements. We have also listed some other approaches using homo-oligomeric nanopores to measure single molecule chemistry activities.

This part has been thoroughly re-written in the revised manuscript and we sincerely apology for the ignorance to discuss other better nanopores. Current societal challenges of how designed reactive site may be introduced to nanopores have been discussed. These discussions were accompanied with references explaining why it is difficult and how it may be solved by PNRSS or similar methods.

Reviewer comments:

-The approach of attaching a streptavidin (or neutravidin...) to the end of the polymer chain (ss DNA or polypeptide) to control the entry of the chimeric molecule and to immobilize this molecule inside the nanopore in order to detect analytes or immobilized chains is not reported in the manuscript introduction. This idea is not new, and the authors should cite previous work involving this kind of system.

Author Response:

First of all, we sincerely apology that relevant citations of previous attempts to attach a streptavidin (or neutravidin) to the end of the polymer chain to control the entry of the chimeric molecule for a static sensing was ignored. We were focusing too much on investigations of nanopore single molecule chemistry and have missed citing these rather significant references. We do apology for our ignorance. In the revised manuscript, these literatures were cited and corresponding discussions were included. We acknowledge reviewer 3 again for picking out the citation issue, which we agree that they are critical.

After thorough search of literatures, we haven't found a previously published work to attach a streptavidin or neutravidin to a piece of polypeptide, as mentioned by the reviewer. **It will be highly acknowledged if reviewer 3 is aware of this reference which describes streptavidin tethered peptide measured in a nanopore sensor and may provide the information to us. PNRSS measurement with tethered polypeptide is very likely a future work following this report. Thus, it will be great to mention this in our discussion of future prospects.** We sincerely acknowledge reviewer 3 again for this suggestion.

Reviewer comments:

-The state-of-the-art is only partially described. Some previous publications are

missing, specifically: the detection of ion binding and the different strategies used for this; and the sensing of transient chemical intermediates, or small species, in engineered nanopores.

Could the authors please add the following references?

Stoddart et al, 2009, Single-nucleotide discrimination in immobilized DNA oligonucleotides with a biological nanopore, PNAS. Add also few other pioneer papers.

Roobahani et al, 2020, Nanopore Detection of Metal Ions: Current Status and Future Directions, small methods. Add also few pioneer papers.

Galenkamp et al, 2020, Directional conformer exchange in dihydrofolate reductase revealed by single-molecule nanopore recordings.

*Bétermier et al, 2020, Single-sulfur atom discrimination of polysulfides with a protein nanopore for improved batteries, Com. Materials.

*Zernia et al, 2020, Current Blockades of Proteins inside Nanopores for Real-Time Metabolome Analysis, ACS Nano.

*Zhou et al, 2018, Monitoring disulfide bonds making and breaking in biological nanopore at the single-molecule level, Science China-Chemistry

Author Response:

We are grateful for the advices to include relevant citations and discussions of corresponding state of the art work in the field. These works have been well cited in the revised manuscript accompanied with other pioneering references and relevant discussions. We thank reviewer 3 again for this suggestion and we do find it quite important to include these materials to merit publication in Nature Nanotechnology.

Reviewer comments:

Results

In general, throughout the manuscript, the authors need to show the limit of detection for the different analytes.

Author Response:

According to the definition of IUPAC and other established reference, the limit of detection is the lowest concentration of an analyte that the analytical process can reliably detect (Anal. Chem. 1980, 52, 2242-2249), according to which the limit of detection of a single molecule sensor is a single molecule. However, I am sure that the reviewer is not concerning that but more about the limit of detection within a reasonable time period. We thus define the limit of detection in this paper as the minimum concentration to detect at least 5 events within a continuous 10 min of measurement. These numbers have been faithfully summarized in **Table S14** in the revised manuscript for your reference.

Please note that the large volume of the measurement chamber and the small size of

our nanopore sensor has resulted in the non-advantageous looking of limit of detection. However, this may be compensated by including more parallel sensors and a flat small measurement chamber such as that reported in a Minlon device, commercially applied to sequence DNA. Some analyte such as vitamin B6, remdesivir, lactic acid and vitamin C may appear in a high enough concentration in natural samples. We thus have demonstrated direct detection of vitamin B6 in human urine sample as a test. We again acknowledge reviewer 3 for the suggestion to include the detection limit. Relevant discussions have also been included in the revised manuscript as well.

Reviewer comments:

How is entry and capture of the PNRSS strand controlled with the traction section,

Author Response:

We thank reviewer 3 for this inquiry. Though the sequence composition of the traction section is not critical, inclusion of a long enough traction section is important to maintain the PNRSS strand in the pore vestibule. Without a traction section, the captured PNRSS strand would spontaneously escape from the pore lumen. This has been well demonstrated with a PNRSS strand without the traction section (**fig. S53**). We again thank reviewer 3 for raising this issue and we realize that inclusion of this discussion is quite helpful for a general audience who may have insufficient experience performing a nanopore measurement.

Reviewer comments:

and what is the main driving force?

Author Response:

Since reviewer 3 didn't specifically mention whether the inquired driving force is for the small molecule analyte or for the PNRSS strand, we will answer this inquiry from both aspects.

The driving force for the PNRSS strand is actually quite clear. Since the original development of nanopore techniques, the electrophoretic forces were considered the main driving force to capture DNA analytes. Some nanopores, including the M2 MspA mutant that we are using also include some positive charges in the pore lumen to trigger generation of electroosmotic flow. Thus, both driving forces are contributing but the electrophoretic force is more dominant.

For small molecule analytes, diffusion, electrophoretic forces and electroosmotic flow of the analyte all contribute and we consider diffusion the main driving force to drive small molecules into the pore lumen. Briefly, placement of the small molecule analyte

to either *cis* or *trans* side of the pore will trigger appearance of events. This is approved by results acquired with norepinephrine placed in *trans* (**fig. S37**) or *cis* (**fig. S57**). The fast diffusion and the small size of the analyte have enabled detection of analyte even if the analyte itself is electrical neutral. Some analytes such as epinephrine, norepinephrine or isoprenaline are electrical positive and could be efficiently driven into the pore by a larger applied potential (**fig. S55**). The contribution of electroosmotic flow seems to be insignificant, as probed by performing PNRSS with an electrical neutral analyte catechol when different voltages were applied (**fig. S54**). These discussions were included in the revised manuscript along with suggested measurements at different potentials.

Reviewer comments:

What is the threshold force to capture the strand for a long time?

Author Response:

We thank reviewer 3 for this inquiry. All through the manuscript, a +160-180 mV potential was applied. Normally a +20 mV potential is enough to maintain the PNRSS strand in the pore for an extended period of time. Application of a high potential is more critical to produce a larger event amplitude when a small molecule was bound to the PRNSS strand so that chemically similar compounds such as epinephrine, norepinephrine and isoprenaline are fully discriminated (**fig. S54-S55**). This is however not achievable when a low potential such as +20 mV was applied. We acknowledge reviewer 3 again for raising this discussion, which is extremely helpful for readers who may lack experiences of nanopore sensing. Other reviewers have also suggested to include measurements at different applied a voltage as well. Relevant discussions of why a potential of +160-180 mV were merged with those added results in the revised manuscript.

Reviewer comments:

Throughout the manuscript, the authors need to provide a description of the data used for the statistical analysis in this study: how many experiments were conducted, how many of these were independent, and how many events were used in the statistical analysis for each experiment.

Author Response:

We sincerely acknowledge the effort of reviewer 3 to improve our manuscript. For each analyte, a minimum of three independent measurements (N=3) were performed to draw a statistical conclusion of its event feature. This has been previously and faithfully mentioned all through the manuscript. The number of events for each measurement has been updated in the legends of corresponding **SI Tables** and corresponding Figure

legends in the revised manuscript as well, following your suggestion.

Reviewer comments:

Figure 1: Did the authors observe a backflow effect of the PNRSS strand due to hydrodynamic interactions?

Author Response:

Thanks reviewer 3 for this inquiry. Backflow of the PNRSS strand was occasionally observed but it was quite rarely happening. Actually, with a continuously applied high potential across the membrane, we can easily maintain the PNRSS strand electrophoretically in the pore lumen for more than 30 min without any escaping of the PNRSS strand. This is more than sufficient to finish most of the measurements as described in this paper. We are grateful for this inquiry and relevant discussions have been added in the revised manuscript just in case other readers may have a similar curiosity.

Reviewer comments:

Figure 1: When the PNRSS strand was not bound (iii) or bound (iv), did the authors observe any difference in the event spikes in terms of current amplitude, current blockade noise, and dwell times?

Author Response:

We thank reviewer 3 for the inquiry. The PNRSS strand when not bound (iii) or bound (iv) with the analyte respectively report two states of sensing. These two states were denoted as I_p and I_b respectively. The difference between I_p and I_b reports the event amplitude ΔI . Or more clearly, when I_p was observed, the PNRSS strand was not bound (iii) and when I_b was observed, the PNRSS strand was bound (iv). We believe that this has been thoroughly discussed already in the original **Figure 1d**. We are pasting this fraction of the figure for your reference as well (**Figure RL3.2**).

Figure RL 3.2. Demonstration of Fig 1d, explaining the event type when the PNRSS strand was not bound (iii) or bound (iv) with the analyte. The levels corresponding to the unbound and the bound state are respectively denoted with red arrows and labels.

Reviewer comments:

Figure 1: The concentration dependence range of Ni²⁺ is in the order of the mM range. What is the minimum concentration needed to observe the interaction?

Author Response:

We thank reviewer 3 for raising this discussion. The minimum concentration of Ni²⁺ to be detected within a reasonable time duration is ~200 nM, as also summarized in **Table S14** of the revised manuscript.

Reviewer comments:

“A first PNRSS strand 13G/14G (Table S1) had two neighbouring guanines that cooperatively bind a Ni²⁺....

Figure 2: What is the experimental evidence for this cooperative interaction?

“Noise fluctuations during Ni²⁺ binding indicate possible reactive intermediates observable by PNRSS”.

Author Response:

We would like to thank reviewer 3 for this inquiry. Regarding results of **Fig. 1**, the cooperative interaction of two neighboring guanines with Ni²⁺ was previously investigated (*Spectrochimica Acta Part A: Molecular and Biomolecular Spectroscopy* **60**, 1907-1915 (2004).; *Journal of Inorganic Biochemistry* **78**, 217-226 (2000).) and predicted by our computer simulation (**fig. S3**) as well. Our experimental results acquired with a single guanine (**fig. S9**) and double guanine (**Fig. 1**) reports different event types (**Table S4**), further supporting the conclusion of previous investigation that the two guanines binding cooperatively with Ni²⁺.

Regarding results of **Fig. 2**, the noise fluctuations observed during Ni²⁺ binding to the triazole of 14TAZ is likely resulted from cooperatively binding of Ni²⁺ to the amine group (**Fig. 2a, b**). It is noticed that the triazole of 14PBA also report binding to Ni²⁺ however the noise has disappeared (**fig. S36 b, c, d**).

Reviewer comments:

Figure 2: It would be interesting to study the noise using Fourier analysis to confirm this hypothesis that reaction intermediates are observable.

Author Response:

Thanks reviewer 3 for this suggestion. We did apply Fourier analysis of corresponding results. However, since the events or noises associated with the generation of reaction intermediates are not uniformly toned in frequency, less information was directly observed by Fourier analysis.

However, for chemical intermediates with a longer dwell time (~ a few ms), it is quite useful to directly capture the newly generated event amplitude, as demonstrated for events of tris acquired at different pH (**fig. S35**). For those highly fluctuating noises that might be associated with the generation of chemical intermediates, it is efficient to analyze their RMS amplitude of the noise when high pass digital filtered. Digital filtration of signal has also applied the concept of spectral analysis and is more suitable to provide useful information for analysis, which is extremely effective in the discrimination of different neutron transmitters (**Fig. 5d, f**) or Remdesivir (**Fig. 6d**).

Reviewer comments:

Figure 2: Why did the authors choose +180 mV applied potential?
“a strand was captured electrophoretically....”

Author Response:

Thanks reviewer 3 for raising this discussion. Technically, a high enough positively applied potential serves to maintain the PNRSS strand in the pore lumen so that PNRSS measurements can be continuously carried out. A higher potential, such as a +180 mV potential, generally maximizes the amplitude of ΔI . For a positively charged analyte in *trans*, such as a Ni^{2+} ion, a higher applied potential also enhances the rate of event appearance. Thus, generally a higher applied potential is more advantageous in the optimization of the sensing performance. However, further increasing of the applied potential may increase the probability of membrane rupture, which is not desired. Consequently, a +180 mV is a suitable magnitude of the applied potential which balances the performance of sensing and to avoid bilayer rupture. Corresponding discussions have been added in the revised manuscript.

Reviewer comments:

Figure 2: Is it only an electrophoresis force or a competition between two other forces such as EOF?

Author Response:

Thanks reviewer 3 for raising this discussion. Analyte detection by PNRSS is a combined effect of diffusion, electrophoresis and EOF. With PNRSS, our major analyte are ions and small molecules, which easily diffuse into the pore lumen to bind with reactive site. Thus, placement of ions or small molecules in either *cis* or *trans* can

trigger the appearance of detection events. The final concentration of the placed analyte and the measurement temperature is critical to guarantee the rate of event appearance (**Table S14**). For charged analyte, such as metal ions or catecholamines, the electrophoresis force further promotes the rate of analyte translocation through the pore (**fig. S55**).

EOF may as well play a role in the modulation of analyte capture however the overall contribution is quite negligible. This is demonstrated when the electrical neutral analyte catechol was tested by PNRSS, from which the reciprocal of the mean inter-event interval ($1/\tau_{on}$) is only slightly modulated by the applied potential (**fig. S54**). It is thus conclusive that the contribution of EOF in this system is rather minimum. Diffusion and electrophoresis are more dominant. The above-mentioned results along with corresponding discussions have been added to the revised manuscript to support our conclusion. We thank reviewer 3 again for raising this discussion, which has significantly improved our understanding of the PNRSS system.

Reviewer comments:

The values of K_b should be compared to published results and interpreted. For example, page 5 line 123, the comparison with NMR and infrared should be determined.

Author Response:

We thank reviewer 3 for the suggestions. Though we are extremely interested to compare the K_b values with results from previous studies. We noticed that references of previous studies using NMR or infrared (*Journal of Inorganic Biochemistry* **78**, 217-226 (2000)) only report information of structures however kinetics information describing exactly the same reaction that we are probing are rarely seen.

We would also like to emphasize that K_b values obtained in different measurement conditions (pH, temperature, salt strength, solvent et al) can vary significantly. It is thus difficult to find literature results measured at exactly the same condition so the comparison described here can only be qualitative. Fortunately, we do find some information related to interactions between PBA and catechol, vitamin C, norepinephrine and epinephrine (**Table S15**).

Some chemical bindings such as ethylene glycol binding to a PBA may be too weak to be detected by conventional assays or may be not of a general research interest to be reported. However, if reviewer 3 is aware of other references of this kind do please let us know so that we can include these comparisons in future revisions.

Reviewer comments:

For the first set of experiments, is it possible to distinguish the different ions in a mixed solution?

Author Response:

Thanks reviewer 3 for raising this discussion. The first set of experiments serve to demonstrate that a combination of natural DNA bases such as A, G or GG can be applied as reactive sites of a PNRSS strand to react with metal ions. We did evaluate the possibility to distinguish different ions in a mixed solution. Although some metal ions such as Co^{2+} , Cu^{2+} did report weak interactions with a PNRSS strand containing a dual guanine reactant, only Ni^{2+} demonstrates a clearly distinguishable interaction, indicating that a combination of natural DNA bases in a PNRSS strand is capable to distinguish between different ions in a mixed solution however the performance is not optimum. It nevertheless could demonstrate the proof of concept of PNRSS by treating GG as the fixed reactant and Ni^{2+} as the mobile reactant. Distinguishing different ion types is better demonstrated with another PNRSS strand 14TAZ, in which the triazole could react with Ni^{2+} or Co^{2+} ions and the binding event is immediately distinguishable (**fig. S17**). We thank reviewer 3 again for raising this discussion. Corresponding discussion has been updated in the revised manuscript as well.

Reviewer comments:

The effect of positive events (page 6 line 184) must be explained and interpreted. If it is not a steric effect, then what could it be? See the example for Tris and the effect of pH compared to pKa values. The same comment applies to all positive events throughout the manuscript.

Author Response:

Thanks reviewer 3 for the inquiry. We, as authors of this manuscript, are also quite interested in this phenomenon of positive events generation. In nanopore research, it is commonly believed that any nanopore blockage would result in negative going events since an analyte would block passage of ions through the nanopore sensor. This generally applies for large sized analyte such as DNA, RNA or protein. However, the analyte that we are investigating are all small molecules. Based on results from many different analytes, it was discovered that all positive going events are associated with boronic acid as the fixed reactant and electrical neutral or electrical negative analyte as mobile reactants. Please note that it is commonly recognized that when a diol complexes with a phenylboronic acid, an anionic boronic ester is formed. This indicates that the charge polarity at the pore constriction is turning negative and might have promoted ion transport through the nanopore, generating an enhanced flow of nanopore current. To summarize, analytes that report positive going events include: tris, catechol, vitamin B6, vitamin C, Remdesivir and others. However, for epinephrine,

norepinephrine and isoprenaline, which are all positive charged analytes and larger in size, negative going events are reported. This has been extremely useful to discriminate structurally similar analyte but has opposite polarities, as demonstrated with catechol and norepinephrine (**fig. S54-S55**).

We speculate that when a piece of ssDNA is trapped inside the nanopore, the overall system is particularly sensitive to the generation of extra negative charge. As observed in another previous publication from our group (*Angew. Chem. Int. Ed.* **2019**, *58*, 8432.), in which a O6-carboxylguanine produces an exceedingly higher pore blockage current than a canonical guanine during nanopore sequencing. Other evidences include that the WT MspA, which has more negative charges around the pore constriction than M2 MspA, has significantly larger open pore conductance. An unpublished result from our group also indicates that introduction of negative charges to pore constriction of M2 MspA would significantly enhance the ionic flow through the pore (**Figure RL 3.3**).

Figure RL 3.3. Enhanced ionic flow of MspA-D than M2 MspA. The asparagine at site 91 of M2 MspA was mutated to aspartic acid, which is a negatively charged amino acid, to generate the mutant MspA-D. The introduced negative charge has significantly enhanced the ionic flow through the pore when a positive potential was applied. A. Scheme of both pore types. B. Traces containing sequential pore insertions from octameric MspA-D. C. I-V curves respectively acquired from both pores. (Unpublished results.)

Many catecholamine derivatives such as dopamine and levodopa may as well be tested. According to this phenomenon, dopamine should report negative going events however levodopa should report positive going events, as estimated from their charge polarities.

Reviewer comments:

What could the intermediate, described on page 7 line 219, be?

Author Response:

The intermediate is a transient state when hydrogen peroxide interacts with a phenylboronic acid. The eventual state is that the phenylboronic acid is irreversibly oxidized to a phenol. Detailed mechanism was discussed and approved in **fig. S36** and **Figure RL 3.4**. Other relevant studies in ensemble were also previously reported (J. Iran. Chem. Soc., 2019: 2379-2388; Eur. J. Org. Chem., 2019: 7307-7321.).

Figure RL 3.4. The description of the intermediate state prior to the irreversible oxidation of a phenylboronic acid to a phenol. Scheme taken from **Fig. 4** of the main text.

Reviewer comments:

Concentrations of hormones and neurotransmitters used in the experiments are of in the of μM range. How relevant are these to physiological concentrations?

Author Response:

Thanks reviewer 3 for raising this discussion. In the revised manuscript, the minimum final concentration to detect an analyte was summarized in **Table S14**, from which a $1\ \mu\text{M}$ final concentration of neuron transmitters is needed to be detectable by PNRSS. According to literatures, the physiological concentrations of neuron transmitters in blood serum is on the order of \sim few nM, indicating that direct placement of blood serum in a PNRSS system will not report enough events within a reasonable amount of measurement time. This is simply due to the fact that only a single nanopore sensor was applied and the detection volume of our measurement chamber is too large ($500\ \mu\text{L}$), which has restricted the detection efficiency. A variety of engineering solutions may be applied to compensate the need. The MinION™ nanopore sequencer device designed by Oxford Nanopore Technology has implemented ~ 2500 active nanopore sensors and each four sensors share an independent channel for signal readout. Simultaneously, their flow chamber is designed to be extremely flat, meaning that the flowed liquid sample will extremely efficiently reach the nanopore sensor for a detection. This state-of-the-art engineering design has thoroughly improved the sensing efficiency however this was not applied in our proof of concept demonstration of PNRSS simply because we don't have access to a specialized Minion device implemented with our nanopore type.

However, we would like to emphasize that some analyte that has been tested by PNRSS do appear in a much higher concentration in real biological samples. These

analytes include Vitamin B6, lactic acid, Remdesivir and Remdesivir metabolite and could be directly detected in a hetero environment. Thus, in the revised manuscript, detection of Vitamin B6 in human urine sample was demonstrated (**fig. S58-59**).

Eventually, we acknowledge reviewer 3 again for raising this discussion. Relevant discussions have been added in the revised manuscript as well. We honestly admit that some analyte such as neuron transmitters do appear in an extremely low concentration physiologically, limiting the detectability directly from a biological sample. However, this may be compensated by an improved hardware containing thousands of equivalent nanopore sensors or after some sample enrichment. We do emphasize that some analytes do appear in a high concentration in a real biological sample and is detectable by PNRSS. Nevertheless, these discussions fall into the engineering aspect of a PNRSS system, likely to be carried out in the near future. These discussions however don't interfere with the demonstration of the core innovation of PNRSS as a proof of concept in this paper.

Reviewer comments:

An interpretation of the delta I observed for each molecule must also be provided. Is it coherent with the size of the molecule, the conformation, and/or the chemical function?

Author Response:

Thanks reviewer 3 for raising this discussion. Generally speaking, the interpretation of the blockage amplitude in a nanopore assay is normally considered to be directly associated with the size of the molecule. This is the case when analytes of extremely different sizes were probed. However, small molecule analytes that differ with only few atoms does not always follow this principle. We believe the size, charge and the conformation all contribute to the overall blockage amplitude. This is however a great topic to be further investigated with plenty of quantum chemistry and molecular dynamics simulation. A brief corresponding discussion was added in the conclusion of the revised manuscript and thanks again for raising the discussion.

Reviewer comments:

The number of events presented in figure S31 seems low. What are the criteria used here for the statistical analysis?

Author Response:

Thanks reviewer 3 for raising this discussion. Results demonstrated in the original **Figure S31** serve to demonstrate how corresponding event types were generated when norepinephrine, epinephrine and isoprenaline were sequentially added. Corresponding investigations have been thoroughly performed in **Fig. 5, fig. S37-42**,

and statistical conclusions were already drawn and the number of events applied to form the statistics has been clearly mentioned in the revised manuscript. To distinguish between different event types, we have applied a support vector classification model (**Fig. 5c**) to automatically recognize different event types and the recognition accuracy has been summarized in the corresponding confusion matrix (**Fig. 5d**). We believe that results in the original **Fig. S31** (now **Fig. S46**) is already clear enough to show the sequential appearance of three populations of event types. However, to perfectly fit the request of reviewer 3, the corresponding measurement was performed again with more extended acquisition time to gain more events for each condition and the results were updated in the revised manuscript for your reference (**Fig. S46**).

Reviewer comments:

Generally, what is the effect of the ionic force, voltage, and temperature on the binding events observed in the manuscript?

Author Response:

Thanks reviewer 3 for raising this discussion. The effect of the ionic force (**fig. S56**), voltage (**fig. S54, S55**) and temperature (**fig. S57**) on analyte binding has been thoroughly discussed by complementing additional experiments.

Briefly, the higher ionic force generates more ions flowing through the pore, thus the event amplitude is generally larger (**fig. S56**). Similarly, the higher an applied voltage brings more ions to flow through the pore as well, thus enhances the event amplitude generally (**fig. S54, S55**). The capture rate of electrically charged analyte is modulated by the applied voltage if the mobile reactant (**fig. S55**). It is however less significant if the mobile reactant is electrical neutral (**fig. S54**). Temperature is critical in the modulation of the reaction rates. Interestingly, PNRSS measurement at ~ 4 °C demonstrates a much extended dwell time of the event and the capture rate is also decreased (**fig. S57**). By complementing these results, the review inquiries should have been well addressed. We again thank reviewer 3 for initiating these discussions, which have significantly improved the research depth of this manuscript. More detailed discussions were included in the section of discussions in the revised manuscript.

Reviewer comments:

Conclusion

The conclusion of the manuscript is too broad. The authors claim that most of the nanopore community's pores could be employed with the PNRSS probe. This is exceedingly hypothetical. Can the authors then show at least one example?

Author Response:

We thank reviewer 3 for this comment. As advised by reviewer 3, a PNRSS measurement using WT α -HL and 14PBA was performed and demonstrated in **fig. S52** of the revised manuscript. The results generally support our conclusion that the PNRSS concept is applicable to other nanopore types as long as a ssDNA containing a suitable fixed reactant can be electrophoretically stretched in the pore lumen. Thus, nanopore types that have been previously applied in studies of ssDNAs should in principle be compatible with PNRSS. We would like to emphasize that different pore geometries when applied to perform PNRSS may affect the sensing performance. In this case, MspA is outperforming WT α -HL by producing a larger event amplitude, more desired to distinguish compounds with similar chemical structures. As also commented by reviewer 3, other nanopores other than α -HL or MspA may demonstrate even better resolution and by including the hypothesis that other nanopore types may also be applied to perform PNRSS, we simply don't want to ignore this possibility and the generality of the PNRSS method itself. So in the revised conclusion, we clearly stated that:

“Nanopores such as cytolysin A (ClyA)¹⁰, fragaceatoxin C (FraC)¹¹, pleurotolysin (PlyA/B)¹² or phi29 Connector¹⁴ with large openings may as well be applied to probe chemical reactions involving larger or more complex mobile reactants, such as polysaccharides or cyclopeptides. However, these proposed plans to apply PNRSS with large channel proteins have not been carried out yet but may be inspiring to other colleagues in the field.”

However, we take the advice of reviewer 3 to be more conservative in corresponding discussions. Since the compatibility of PNRSS with solid state nanopores and DNA nanopores have not been tested, we decide to avoid acclaiming this possibility, as suggested by reviewer 3. However, we would be excited to see if researchers with expertise in solid state nanopores or DNA pores may find PNRSS useful. Please note that nanopore single molecule chemistry has never been successfully carried out in any solid state nanopore setup largely due to the poor resolution. PNRSS may be inspiring for these approaches.

I am not sure whether reviewer 3 find any other conclusions to be too broad. To avoid any miscommunication which may delay the publication, the conclusion has been segmented and responded item by item for your reference. We believe that all conclusions are well supported by concrete experimental evidences provided in this manuscript.

Conclusion 1: To summarize, PNRSS, first reported in this paper, serves as a convenient molecular toolkit with which to study single molecule chemistry processes with a nanopore. The specific aim is to break the technical bottleneck to introduction of any number or type of reactive groups into any spot of the nanopore lumen, which is difficult, even for the best in the field. With PNRSS, however, this difficulty has been

transformed instead to synthesis of functional DNA oligomers, a routine performed daily by countless biochemistry labs or as a low cost service provided by a variety of commercial vendors.”

Discussion 1: We believe that this part of the conclusion is well supported by all materials in this manuscript. The concept of PNRSS has never been previously published. It also demonstrates the easiest and the most economic solution for any lab to perform nanopore based single molecule chemistry observations.

Conclusion 2: Distinct from the previous configuration, PNRSS has also enabled repetitive monitoring of irreversible reactions, further broadening the choice of chemical reactions that previously were difficult to study by a nanopore.

Discussion 2: This part of conclusion is supported by materials in **Figure 3**. It is also a fact that repetitive monitoring of irreversible reactions can't be carried out by engineered pores with fixed reactive sites, which can only monitor such type of reactions once.

Conclusion 3: We report a sum of 20 single molecule chemical reactions, in which hydrogen peroxide, buffer reagent, transition metal ions, glycerol, lactic acid, vitamins, catecholamine derivatives or anti-viral medicines participate.

Discussion 3: This part of conclusion is supported by major part of this manuscript, which were also summarized in **Table S16**.

Conclusion 4: Though limited by the length of this paper, these reactions demonstrate core aspects of the PNRSS technique, and show its feasibility and versatility. To the best of our knowledge, this is the largest number of nanopore based single molecule chemistry reactions that have been reported in a single publication. Most of them have never been previously investigated as single molecules (**Table S16**).

Discussion 4: We do believe that this manuscript demonstrates the largest number of nanopore based single molecule chemistry reactions that have been reported in a single publication, acknowledging the programmable nature of PNRSS.

Conclusion 5: Given the conical structure of MspA, which efficiently focuses ionic flows to its narrow pore restriction, all single molecule chemical reactions discussed in this paper have demonstrated a fully resolved event amplitude.

Discussion 5: This is supported by the large amplitude of most single molecule chemistry event produced by this system. It is also supported by the comparison between MspA (**Figure 5b**) and WT α -HL (**fig. S52**) when isoprenaline was identically applied as the mobile analyte.

Conclusion 6: For PBA, direct recognition of epinephrine, norepinephrine and isoprenaline also suggests its immediate biomedical applications.

Discussion 6: This is well supported by results in **Figure 5**.

Conclusion 7: The single molecule discrimination of remdesivir and its triphosphate metabolite by PNRSS may inspire pharmacokinetics measurements with a variety of nucleoside analogue medicines.

Discussion 7: This is well supported by results in **Figure 6**.

Reviewer comments:

Why did the authors use MspA and not alpha-hemolysin for instance?

Author Response:

Thanks reviewer 3 for the comment. MspA demonstrates a more optimum geometrical advantage than WT α -HL by generating a larger event amplitude, which is more desired to discriminate between compounds that are similar in the chemical structure. We believe that it is well supported by the added experiment when PNRSS was performed with WT α -HL (**fig. S52, Figure RL 3.1**), which was kindly suggested by you. We do acknowledge your kind suggestion which has significantly improved our manuscript.

By the end, we believe all above comments from you have been well and fully addressed. Your suggestion to perform PNRSS using WT α -HL is fantastic. To ensure the reproducibility of PNRSS measurement using either MspA or WT α -HL, plasmids corresponding to both pore types were shared in the public repository as well.

Access link: <https://www.molecularcloud.org/s/shuo-huang>

Access codes: **M2 MspA:** MC_0101191; **α -HL WT:** MC_0068416;

After three decades of development, the nanopore community is getting bigger and bigger, more like a family. By openly providing these plasmids, I do hope my humble effort and the invention of PNRSS can inspire new insights in the field.

Your other suggestions to discuss the EOF, electrophoretic force, ionic strength, temperature and the voltage on the performance of PNRSS is also extremely constructive, which have thoroughly improved our understanding of this new system. We sincerely feel fortunate to have you as our reviewer and as a junior research group in the field we do treasure this opportunity that this work may eventually be published by such a high impact journal. **I apology that the response letter is a bit long. As a junior research group in the field, I do treasure this valuable opportunity that my**

proud invention may be published in such high impact journal. Thus, I try my best not to miss any point in this response letter. I sincerely acknowledge your

Figure RL3.5. An artistic demonstration of PNRSS. PNRSS is a highly modular technique. It is like a tool box, by combining different reactive modules, a large variety of chemical sensing function could be enabled. The more difficult protein engineering has been omitted. Please also don't miss watching Video S7 for an animation describing the PNRSS method.

time reading to this end and your support to publish this work will be highly appreciated. Actually, I am quite proud of this rather systematic and useful invention which may be useful to many colleagues in the field. To better illustrate the idea of PNRSS and to draw attention of a broad audience, an artistic illustration (**Figure RL3.5**) has also been provided to the editorial team. I understand that most reviewers of Nature Nanotechnology must be very busy, I however strongly suggest reviewer 3 to pay attention to all accompanied Supporting Video files, which might be useful to evaluate our data quality. An artistic animation is as well provided as Video S7 to illustrate the concept of PNRSS as well.

Reviewer #4 (Comments for the Author):

Reviewer comments:

This is an article that illustrates a the nanopore-based design for probing single-molecule stochastic sensing of small-organic molecule reactions in real time. The overall concept of stochastic sensing of small-molecule reactions is no longer new and it has been well paved by authors' predecessors, including those cited in the manuscript. The overall goal of this work is obtaining a generic sensor for a multitude of small-molecule reactions. Numerous data are included in the main text and supporting information. However, this reviewer raises serious issues regarding to quantitative aspects of the presentation, as follows.

Author Response:

First of all, we would like to thank reviewer 4 for recognizing the core innovation of this manuscript, which is to obtain a generic nanopore sensor for a multitude of small-molecule reactions. We also acknowledge reviewer 4 for recognizing that relevant citations from our predecessors have been well cited. However, we would like to emphasize that though nanopore single molecule chemistry was previously established by our predecessors, the main difficulty still lies on how to engineer a nanopore to have a singly active reactive site right at an appropriate location in the pore lumen. The most investigated nanopore type for previous single molecule chemistry observation is α -HL. Being an heptameric porin, engineering of α -HL to be a heterooligomeric porin containing a singly altered monomer however maintaining the remaining six monomers unaltered is not a trivial task at all. **It is because of this difficulty, report of a heterooligomeric porin to investigate a single molecule reaction has never been reported by any group other than Prof. Hagan Bayley's, to the best of our knowledge. It is hard to imagine that such a powerful technique has not yet been thoroughly mastered by any other groups in the field, indicating the importance to break this technical hurdle.**

We would also like to emphasize that these reports including ours (ACS Nano 2018, 12, 1, 786–794; ACS Nano 2019, 13, 4, 4101–4110; Nature Communications volume 10, 5668 (2019); Science China Chemistry volume 61, pages1385–1388 (2018)) are not yet based on successful engineering of a heterooligomeric porin and the demonstrated performance of these homooligomeric porins is far from comparable to the corresponding heterooligomeric counterparts. Simultaneous appearance of events with multiple steps are observed, which is not desired for analysis (**Figure RL 4.1**) but can't be easily avoided if without exceptional purification techniques.

Borsley S, Acs Nano 2018, 12(1): 786-794.

Haugland MM, Acs Nano 2019, 13(4): 4101-4110.

Zhou B, Science China Chemistry 2018, 61(11): 1385-1388.

Cao J, Nat Commun 2019, 10(1): 5668.

Figure RL 4.1 A demonstration of single molecule chemistry measurements acquired from homo-oligomeric nanopore sensors. Multiple steps of events corresponding of simultaneous reactions from multiple reactants were observed.

With PNRSS however, this difficulty is easily solved by generating a synthetic ssDNA containing the desired reactive site. The composition and the location of the reactive site can be easily adjusted by designing different ssDNAs so that new functions of small molecule sensing are immediately enabled. Engineering of a biological nanopore is actually not required at all. **Thus, we believe that the biggest achievement of this manuscript is that a long-standing technical hurdle has been broken and the solution is rather straightforward and highly feasible and can be mastered by any group in the field.** Another reviewer (reviewer 3) has suggested to apply PNRSS in a different biological pore and it has also been demonstrated with WT α -HL in the revised manuscript, confirming the ease and the generality of this new method. Acknowledging this gained programmability and versatile nature of PNRSS, numerous data are included to support our discussions, as also recognized by reviewer 4 as well. An artistic picture below (**Figure RL 4.2**) well describes the core innovation of our work. **Briefly, to enable numerous types of single molecule sensing functions, modularly designed synthetic polymer was applied as components of a toolbox. To the best of our knowledge, this concept has never been previously reported, which forms the core innovation of this work.**

Besides, the application of MspA has thoroughly optimized the sensing performance

of PNRSS by producing much enlarged event amplitude so that epinephrine, norepinephrine and isoprenaline can be directly distinguished. A COVID-19 drug Remdesivir and its metabolite were also distinguished as well, suggesting an extremely high resolution of sensing. As suggested by another reviewer (reviewer 3), a side by side comparison using α -HL was included in the revised manuscript as well, further confirming our hypothesis that the application of MspA is a big leap in the advancement of this technique.

Figure RL 4.2 An artistic illustration of the concept of PNRSS. Please also don't miss watching Video S7 for an animation describing the PNRSS method.

We also acknowledge reviewer 4 for pointing out some quantitative issues on the demonstration of corresponding histograms concerning the time constants. **Actually, in our previous manuscript, these quantitative numbers were directly applied to derive the mean interevent intervals and the mean dwell times. Other quantitative numbers such as the K_d values were already included in the original manuscript but placed in the SI.** These issues have all been well addressed in the revised manuscript.

Reviewer comments:

Major concerns.

-Unfortunately, the authors do not provide standard inter-event and dwell time histograms that are representative to their end-points or consolidated plots. For example, most figures show simple bimolecular association processes and unimolecular dissociation events (e.g., Fig. 1f), yet none of them demonstrate these

facts via direct fits of primary event data. In other words, the manuscript fails to show a standard event analysis that would realistically lead to reliable numbers. Instead, scatter plots are provided (e.g., Fig. 1e), although these are less informative than standard time-constant histograms. Furthermore, these histograms' fits should be used to verify that the end-point graphs are indeed correct. These histograms should also show whether time constant distributions of dissociation events follow a single- or a multiple-exponential probability function.

Author Response:

We thank reviewer 4 for the comments. The histogram concerning the reaction time constants were previously used to derive the mean inter-event interval and the mean dwell time. To save space, the histograms were not demonstrated item by item. However, we would like to emphasize that our original **fig. S4** (now **fig. S4**) have clearly explained how these time constants were derived and histograms with single exponential fitting were definitely previously shown (**Figure RL 4.2**). As requested by reviewer 4, all time histograms were included in the revised manuscript and this issue should have now been well addressed. We apology for not being able to guide reviewer 4 to corresponding **SI** items in our previous manuscript. As requested by reviewer 4, standard inter event interval and dwell time histograms were also provided in the revised manuscript.

Reviewer comments:

-Methods section on nanopore measurements only discusses how the median currents were determined. There was no information on how time constants were determined. Because time constants are pivotal part of this detection analysis, this information is essential to check that the finalized plots reflect the event distribution and event duration on corresponding electrical traces. Also, the authors should choose a time scale that allows the reader to compare durations of events and those from fits of dwell-time histograms, as well as their corresponding event frequencies.

Author Response:

We thank reviewer 4 for raising this discussion. The determination of major event parameters was thoroughly described in the original **fig S4**. Demonstration of histograms of t_{off} and t_{on} along with single exponential fitting results were as well shown. We speculate that reviewer 4 may have missed this part and we are pasting the original **fig S4** here for your reference (**Figure RL4.2**). We would also like to emphasize that other than the time constants, event features such as the mean blockage depth, the event noise, the event noise after treatment of a high pass filter are also critical to distinguish between chemical compounds with minor structural variations.

All traces or events were objectively demonstrated with suitable time scales to show

details of event features, which should be enough to compare the duration of the event dwell time and the frequency of the event appearance whereas other event features such as fluctuations of the blockage level are still clearly seen. If reviewer 4 has any suggestion to change any particular time scale, we would be more than happy to revise the time scale as requested.

Figure RL 4.2. Demonstration of the original **fig S4**. Representative histogram containing single exponential fitting (**c, d**) was already included. Determination of time constants was also well described.

We find it more objective and quantitative to compare the duration of the event dwell time and the frequency of event appearance more quantitatively from the derived τ_{on} and τ_{off} values in the corresponding SI tables. The τ_{off} values have been summarized in **Table S4**, **Table S6**, **Table S8** and **Table S11** for your reference. The τ_{on} values were applied to derive the k_{on} values as summarized in **Table S9**, **Table S10**, **Table S13**. We do apology for not being able to guide reviewer 4 to these corresponding **SI** items. We have correspondingly revised the manuscript by including corresponding time histograms in the SI, as requested by reviewer 4. We believe the above discussions should have well addressed the inquiries of reviewer 4 and we acknowledge reviewer 4 again for these suggestions.

Reviewer comments:

-Figs. 1, 3, 5, and 6. Here, standard histograms of dwell times and interevent intervals should be provided. For example, in Fig 3, on the right side only one event is shown. This is not acceptable for a process that requires intensive data statistics to arrive at reliable outcomes.

Author Response:

We thank reviewer 4 for raising this discussion. In the revised manuscript, all standard histograms of dwell times and interevent intervals are provided as requested however in the SI. These correspond to **fig. S5**, **fig. S8**, **fig. S10**, **fig. S14**, **fig. S16**, **fig. S22**, **fig. S24**, **fig. S26**, **fig. S28**, **fig. S30**, **fig. S32**, **fig. S34**, **fig. S38**, **fig. S40**, **fig. S42**, **fig. S48** and **fig. S50**. These results were previously acquired to derive the τ_{on} and

τ_{off} values against different analyte concentrations.

As you can see now in the revised manuscript, there are too many histograms of this kind to be included in the main text. We apology for not being able to demonstrate these results in the main text figure. It was to avoid distracting readers to recognize the true core innovation of this work, which is how PNRSS was carried out. However, these results can definitely be provided in the SI of the revised manuscript.

The following issue is again due to our ignorance for not being able to guide reviewer 4 to corresponding SI items. However, we would like to emphasize that the requested materials were already provided in the original submission.

“in Fig 3, on the right side only one event is shown. This is not acceptable for a process that requires intensive data statistics to arrive at reliable outcomes.”

Figure RL 4.3 Demonstration of the original Fig 3 and its corresponding supportive content in the original SI. As suggested by reviewer 4, histogram of event times along with single exponential fittings were as well included. Due to a limited space here, only items corresponding to discussion of Catechol detection is provided in this reference letter figure. We would like to emphasize that for each new compound tested in this paper, a set of corresponding supportive content was included in the SI.

We would like to emphasize that in **Fig 3**, the demonstration of a single event for each analyte represent the conclusion that this event is a representative event corresponding to this measurement. A thorough evaluation of each analyte has been performed by a minimum of three independent trials (N=3) in our original submission. These have been intensively studied and demonstrated with results of representative traces, derived time constants at various analyte concentrations and scatter plot of ΔI vs τ_{off} . I am pasting one example of these results for your reference (**Figure RL 4.3**).

Since we have provided studies of numerous analyte to demonstrate the programmability and the generality of this newly developed method. It is rather difficult to place all details in the main text figures. But they were faithfully included in the SI of our original submission. We apology again that our original manuscript didn't lead the reader to corresponding SI items. These have been improved by clearly guiding readers to read corresponding SI items and the time histograms of the revised manuscript (see the figure legend of **Fig. 3**). The suggested stand histograms of dwell times and interevent intervals were as well included in the revised manuscript. We acknowledge reviewer 4 again for your valuable comments.

Reviewer comments:

-Many of these reactants are (e.g., metal ions) are charged. Therefore, it is conceivable that their binding interactions with the specific reactive site is voltage dependent. It is not clear what is the departure of measured/calibrated quantities from those expected in bulk solution.

Author Response:

We acknowledge reviewer 4 for raising this discussion. The reviewer is correct that many of these reactants are charged and a differently applied potential does modulate its capture rate. As suggested by another reviewer (reviewer 3), measurements with representative electrical neutral molecules (catechol, **fig. S54**) and positively charged molecule (norepinephrine, **fig. S55**) were tested, from which it is conclusive that the mean dwell time (τ_{off}) is almost independent of the applied voltage but is dependent of the type of the reaction. However, if the analyte is charged the on time is linearly dependent of the applied voltage.

Chemical reactions are extremely sensitive to change of conditions, such as temperature, pH, salt strength and so on. It is hard to find results from previously reported literatures describing kinetics values performed at exactly the same condition as we have performed by PNRSS. Still, we find some results for a comparison (**Table S15**). The derived K_b values when compared with quantities measured in bulk is actually pretty close to each other and has been mentioned in the section of limitations of the revised manuscript. However, we would like to emphasize that the main purpose to perform PNRSS measurement is not to measure the K_b values corresponding to that expected in bulk. It is actually the existing knowledge of chemical reactions in bulk which has inspired the design of the single molecule sensing assay in a nanopore and we have demonstrated 20 cases of them in this paper, acknowledging the high flexibility of measurement design by the new PNRSS method. Still, we would like to thank reviewer 4 for raising this discussion. Relevant results and discussions have been added in the revised manuscript accordingly as well.

Reviewer comments:

-There is no discussion on quantitative values like K_d . Are these values expected? What potential mechanisms exist, if obtained values deviate from expectations.

Author Response:

We acknowledge reviewer 4 for raising this discussion. First of all, the derived quantitative values like K_d , which is the reciprocal of K_b , has been previously provided in the original manuscript however in corresponding **SI Tables**. **We again apology for not being able to guide reviewer 4 to read corresponding SI items, though the requested materials were definitely provided in the original submission.** For your reference, the original **Table S9** has been pasted below (**Table RL 4.1**). The K_b values have been highlighted in the table as well. Please note that the K_b values of all reactions tested in this paper have been provided in the original submission but in the SI.

Table RL 4.1 | Kinetic constants for the formation of complexes between PBA and diols.

The PNRSS measurements were performed as described in **fig. S21-32**. A buffer of 1.5 M KCl, 10 mM HEPES, pH 8.0 was used. A +160 mV potential was continuously applied. For each measurement, kinetic constant values were derived from events within a 15 min continuously recorded trace. Three independent measurements (N=3) were performed for each condition to produce the statistics. (This is the original Table S9 in our previous submission.)

Mobile Reactants	k_{on} ($M^{-1}S^{-1}$)	k_{off} (S^{-1})	K_b (M^{-1})
Catechol	1000±300	0.98±0.08	1100±300
Ethylene glycol	43±6	390±40	0.11±0.01
Glycerol	99±8	76.2±1.8	1.30±0.08
L-Lactic acid	260±70	64±4	4.1±1.3
Vitamin C	320±70	530±50	0.60±0.12
Vitamin B6	57580±9160	23.9±0.8	2410±400

These values are expected. The general rule of binding between reactants is exactly expected in a way that strong interactions in bulk were probed also as strong interactions in single molecule as well. It is the existing knowledge of chemical reactions in bulk which has inspired the design of the single molecule sensing assay in this paper.

It is rather normal to see the derived K_d or K_b value to be slightly off from that measured in bulk and reported in literatures since they were measured in a different conditions. It is hard to find previously reported measurements in bulk adapting exactly the same measurement condition. Still, based on values adapted from the literatures, the derived K_b generally fits our expectation that stronger interactions measured in bulk generally report larger K_b values when measured in a nanopore. These comparisons are summarized in **Table S15** of the revised manuscript. Relevant discussions were also added in the section of limitations in the maintext. We again acknowledge reviewer 4

for pointing this out, we have added corresponding discussions in the revised manuscript to explain this.

Reviewer comments:

-Authors should make efforts to discuss and compare their approach with other competing technologies.

Author Response:

We would like to thank reviewer 4 for this suggestion. It is definitely important to include these discussions. The 2nd paragraph of the revised manuscript has been expanded to include more discussions of the state-of-the-art and competing technologies using engineered nanopores to revolve binding of single molecules. The 3rd paragraph explains why PNRSS is a suitable technique to address these challenges and why it might be advantageous and more flexible in the design. We again acknowledge reviewer 4 for this valuable suggestion to improve the writing of the introduction.

Reviewer comments:

-These are experiments when the analytes are added in a clean (homogeneous) solution. It is not clear how this stochastic sensing becomes quantitative in a heterogeneous solution, which is otherwise nearer to a realistic environment. This is because the entire analysis is based on current blockages. How do the constituents of a heterogeneous solution interfere with the targeted events produced by small-molecule analytes? The authors should provide experimental evidence for a proof-of-concept that they can perform these experiments with quantitative reliability in a heterogeneous solution, such as in other state-of-the-art nanopore sensing formulations.1-2

REFERENCES

1. Galenkamp, N. S.; Soskine, M.; Hermans, J.; Wloka, C.; Maglia, G., Direct electrical quantification of glucose and asparagine from bodily fluids using nanopores. *Nature communications* 2018, 9 (1), 4085.
2. Fahie, M. A.; Yang, B.; Mullis, M.; Holden, M. A.; Chen, M., Selective Detection of Protein Homologues in Serum Using an OmpG Nanopore. *Analytical chemistry* 2015, 87 (21), 11143-11149.

Author Response:

Thanks reviewer 4 for raising this discussion and the suggestion to perform at least one demonstration of PNRSS in a heterogeneous solution. Actually, we do find it an inspiring suggestion, which demonstrate the specificity of this measurement against

interfering analyte in a heterogeneous solution. Events of vitamin B6 has a unique event pattern, which is highly recognizable. True human urine sample is also the easiest heterogeneous sample that we can access. Physiologically, vitamin B6 does appear in human urine sample in a high concentration. To mimic this situation, corresponding PNRSS measurements were designed and performed (**fig S58-59**). **Actually, true human urine samples used in this study was kindly donated by the first author Wendong Jia (asian, male, age ~27)**. The results also demonstrate a clear concentration dependence, confirming a quantitative reliability when measured in a heterogeneous solution.

This should have well addressed the suggestion provided by reviewer 4. We do find this suggestion quite innovative, which has thoroughly improved the quality of this manuscript. We would like to acknowledge reviewer 4 for such a constructive idea, which has seriously inspired some future insights that this system could be practically applied.

Figure RL 4.4 Demonstration of vitamin B6 detection in human urine. The results are pasted here for your reference. Please refer to fig. S58-59 for other details of the measurements.

By the end, it is fortunate to have reviewer 4 to comment on this manuscript. We are sorry that some SI items that you are interested were not noticed by you, which we can simply address them by rewriting corresponding sections of this manuscript. We sincerely hope that reviewer 4 has now recognized the urgency to develop a highly programmable reactive nanopore sensor and how difficult it can be achieved by

previously reported strategies. Besides, for the purpose of technology validation and exchanging, the core plasmid DNAs coding for WT α HL and M2 MspA are also openly shared in the plasmid repository as well. We do hope our humble effort would stimulate more insights in the field of nanopore research by enabling more researches in this direction.

Access link: <https://www.molecularcloud.org/s/shuo-huang>

Access codes: **M2 MspA**: MC_0101191; **α -HL WT**: MC_0068416;

REVIEWERS' COMMENTS

Reviewer #5 (Remarks to the Author):

The manuscript reports a new chemical sensing strategy based on the nano-reactors built in the protein nanopore inner-channel. I have carefully read the manuscript and the responses to the previous 4 reviewers. I tend to agree that after revision, the author have carefully addressed most concerns raised the reviewers. Now the manuscript is well-presented in preparation, interesting in concept and convincing in results, which makes it being suitable to Nature Communications.

I support the manuscript mainly in two star points. First, I agree with the reviewer 1 that there are pioneer publications having reported the nano-reactors using nanopore technique. Even though, in this manuscript, the smart and innovative design of a single PNRSS strand enables the nano-reactor to be easier-fabricating, more programmable and general to many types of chemicals. This effort obviously decreases the technique and cost barrier encountered by earlier publications, and finally brings clear application possibilities of a nanopore other than merely sequencing. Second, the signal-to-background ratio for a target chemical is high. This is a very important advance to provide convincing and reliable sensing signals.

The comments of previous 4 reviewers are professional and comprehensive, which led to very significant improvements of the manuscript. Therefore, I didn't find additional concerns in principle, data or conclusions. There are merely small questions left to improve the discussion.

1) The relatively low sensitivity (lowest detection concentration) may be a concern. I don't anticipate this single paper can solve all the practical issues for molecular detection. However, necessary discussion on why and how the sensitivity should be improved needs to be added. Is it because the limited translocation rate? And how can the translocation rate be improved? Will there be bias in the translocation rates among different targeting chemicals? Will the bias affect the analogue discrimination? How to solve these concerns?

2) The manuscript should roughly define the target/molecule range (sizes, molecular weights, bond types, functional groups...?) and sample sources (urine, blood, tissue, drug tablet, etc...?) that may be detected by the PNRSS sensor.

3) The PNRSS is more like a nano-sensor than nano-reactor. In my opinion, the function of a reactor is to prepare/produce a new compound/specie, rather than signaling. I am not sure whether the emphasis of the "reactor" may confuse some readers. Therefore, I suggest, if really necessary, relative changes be made in the title or certain description.

Reviewer #6 (Remarks to the Author):

The manuscript "Programmable Nano-Reactors for Stochastic Sensing" by Wendong Jia et al, studies several chemical reactions, for up to 20 single molecules, using a protein nanopore. To answer this challenge, the authors designed and manufactured an elegant nanoreactor with a MspA nanopore used to dock a streptavidin tethered PNRSS strand. A part of this strand contains a versatile reaction section with different kinds of reactive sites to perform the chemical reactions. This work's central claim is shown: the ability to use the nanoreactor as a versatile platform to measure the association/dissociation rates and binding constants of different chemical reactions at the single-molecule level. Another interesting claim is the ability to detect chemical intermediates.

All the comments made in the previous review were well answered along with appropriate

modifications in the manuscript. Furthermore, I would like to emphasize that new experiments were achieved showing this system can be universally used with protein nanopores, demonstrated here with an alpha-hemolysin pore. This new finding clearly increases the significance of the study elevating it to a substantially valuable contribution to the field. Additionally, a number of 20 molecules were analyzed in this paper while most of the nanopore community usually focuses on just a few molecules. This is an excellent work and this is why I strongly recommend this paper for publication in Nature Communication.

Comment to the authors:

"The approach of attaching a streptavidin (or neutravidin...) to the end of the polymer chain (ss DNA or polypeptide) to control the entry of the chimeric molecule and to immobilize this molecule inside the nanopore in order to detect analytes or immobilized chains is not reported in the manuscript introduction. This idea is not new, and the authors should cite previous work involving this kind of system"

From the answer given by the authors I believe the comment was maybe not clear regarding chimeric molecules trapped inside nanopores. Here are some examples of this approach already published (non-exhaustive list):

Pastoriza et al, ACS Nano, 2014 ; Rosen et al, Nat. Biotech, 2014 ; Rodriguez Larrea et al, Nat. Nanotech, 2013 ; Rodriguez Larrea et al, Nat. Comm., 2014 ; Nivala et al, ACS Nano, 2014. ; Nivala et al, Nat. Biotech., 2013

Reviewer #7 (Remarks to the Author):

This thorough manuscript revision has increased the strength and readability of this work. The authors included the standard inter-event and dwell time histograms which adequately addresses one of the major concerns raised. Also the newly introduced detection of vitamin B6 in urine significantly strengthens the potential applications for PNRSS. I feel the authors have updated the manuscript in a way that most of reviewer's reservations have been remedied but a few concern can still be rectified.

The concerns raised about how time constants were determined can still be addressed. The authors should draw attention to Fig. S4 earlier in the "Nanopore measurements and data analysis" section if they want to use this figure as their only explanation of how they are determined. Also the trace time scales can still be altered to allow the reader to visually confirm the *t_{off}* and *t_{on}* values being reported. I understand that the authors are using Fig. 1d as an example trace but it would be helpful if they mentioned the concentration of Ni²⁺. This will help the reader confirm the presented data for themselves. The authors can replace the 500 ms axis with 100 ms in the highlighted region of the trace so the ~12 ms dwell time events are more obvious.

The authors strength the paper by introducing material showing PNRSS detection of neutral and positively charge analytes. Both the neutral catechol and positive norepinephrine provide critical data showing that the mean dwell times are independent of voltage. It is not clear why the association of the charged analyte is linearly dependent on the applied voltage. I would expect it to be exponentially dependent based on the equation $k_{on} = A(0) e^{(-q\Delta U/(kBT))}$. The authors should discuss why they observe a linear voltage dependence.

This version of the manuscript better assists the reader to supplementary information which helps address a lot of the previous concerns. Therefore, I feel the authors have satisfactorily addressed the concerns raised by reviewer # 4.

Response Letter

Reviewer #5 (Remarks to the Author):

Reviewer Comments:

The manuscript reports a new chemical sensing strategy based on the nano-reactors built in the protein nanopore inner-channel. I have carefully read the manuscript and the responses to the previous 4 reviewers. I tend to agree that after revision, the author have carefully addressed most concerns raised the reviewers. Now the manuscript is well-presented in preparation, interesting in concept and convincing in results, which makes it being suitable to Nature Communications.

I support the manuscript mainly in two star points. First, I agree with the reviewer 1 that there are pioneer publications having reported the nano-reactors using nanopore technique. Even though, in this manuscript, the smart and innovative design of a single PNRSS strand enables the nano-reactor to be easier-fabricating, more programmable and general to many types of chemicals. This effort obviously decreases the technique and cost barrier encountered by earlier publications, and finally brings clear application possibilities of a nanopore other than merely sequencing. Second, the signal-to-background ratio for a target chemical is high. This is a very important advance to provide convincing and reliable sensing signals. The comments of previous 4 reviewers are professional and comprehensive, which led to very significant improvements of the manuscript. Therefore, I didn't find additional concerns in principle, data or conclusions. There are merely small questions left to improve the discussion.

Author Response:

We highly acknowledge reviewer 5 for a precise description of the main topic of this manuscript and also for recognizing the core innovation of the PNRSS strategy. We also acknowledge your comments on remaining minor issues to improve the discussion. These comments are addressed below in a point by point style for the reference of you and the editorial team.

Reviewer Comments:

1) The relatively low sensitivity (lowest detection concentration) may be a concern. I don't anticipate this single paper can solve all the practical issues for molecular detection. However, necessary discussion on why and how the sensitivity should be improved needs to be added. Is it because the limited translocation rate? And how can the translocation rate be improved? Will there be bias in the translocation rates among different targeting chemicals? Will the bias affect the analogue discrimination? How to solve these concerns?

Author Response:

We acknowledge reviewer 5 for raising such valuable discussions. We agree with reviewer 5 that the current lowest detection concentration of the analyte is not yet satisfying. Even though, we have demonstrated direct sensing of VB6 in real human urine samples using this newly developed strategy.

I think the current limitation of the sensitivity is restricted by the intrinsic reaction rates of the analyte with the reactive site of PNRSS, meaning that even a significant amount of analyte has pass through the pore, only a fraction of them would bind to the reactive site to be detected. To improve the performance, we can either have more parallel nanopore sensors in an array, such as those demonstrated by Oxford Nanopore Technologies, to improve the efficiency of detection. Alternatively, we may include multiple reactive site or an optimized reactive site which may bind the analyte more efficiently. However, considering that this is the first paper of its kind and the detection of VB6 from real human urine sample has been demonstrated, further optimization can be performed in a series of follow up works. Relevant discussions were already included in the manuscript. We are pasting the discussions here for the ease of your references as well.

"Due to the large volume of the measurement chamber and the small size of a single nanopore sensor, the efficiency of detection is generally not optimum for the current configuration. Thus, without any sample enrichment, it is not expected to observe enough events for quantification when a target analyte was present in a low concentration in a physiological sample. This is the case for epinephrine, norepinephrine and isoprenaline which have a \sim nM physiological concentration in blood serum however the detection limit reported here is \sim 1 μ M. Engineering wise, this may be improved by introducing thousands of independent parallel nanopore sensors and an extremely flat flow cell to boost the sensing efficiency, similar to a configuration demonstrated in a MinION sequencer. On the other side, some analyte tested in this paper may appear at a high concentration in natural samples and may be directly applied for detection even with the current setup."

Reviewer Comments:

2) The manuscript should roughly define the target/molecule range (sizes, molecular weights, bond types, functional groups...?) and sample sources (urine, blood, tissue, drug tablet, etc...?) that may be detected by the PNRSS sensor.

Author Response:

We thank reviewer 5 for the suggestion. I think the Supplementary Table 16 should well summarize the target molecule range. All chemical structures, bond

types and functional groups are given and highlighted. All tested chemicals are pure synthetic compounds, tested as the standard. We are pasting the relevant discussions below for your reference as well.

*"To the best of our knowledge, this is the largest number of nanopore based single molecule chemistry reactions that have been reported in a single publication. Most of them have never been previously investigated as single molecules, as summarized in **Supplementary Table 16.**"*

Reviewer Comments:

3) The PNRSS is more like a nano-sensor than nano-reactor. In my opinion, the function of a reactor is to prepare/produce a new compound/specie, rather than signaling. I am not sure whether the emphasis of the "reactor" may confuse some readers. Therefore, I suggest, if really necessary, relative changes be made in the title or certain description.

Author Response:

We acknowledge reviewer 5 for the comment. We agree with reviewer 5 that the purpose of PNRSS is not to generate new compound but for single molecule sensing. For this purpose, the term PNRSS stands for "Programmable Nano-Reactors **for Stochastic Sensing**". We definitely have defined that this reactor is for sensing instead of manufacturing. Considering that a single molecule chemical reaction is definitely occurring, which is critical for sensing, we believe the term PNRSS is well describing the nature and the purpose of this technique and can be discriminated from other nanopore sensors in which no chemical reaction is involved. Besides, the concept of nanopore single molecule reactor was historically initiated by Prof. Hagan Bayley from University of Oxford, as you can see from these literatures (**Nature Biotechnology** 18, 1005-1007 (2000).; **Angew. Chem. Int. Ed.** 2020; 59(36):15711-15716; **Science**, 361, 908 (2018).). We are thus extremely reluctant to give it a different name. However, a brief description to remind the reader that this nanoreactor is intentionally designed for sensing has been added in the revised manuscript. We are pasting this part of discussion below for your reference.

"Results in this manuscript might be inspiring in preparation of chemical compounds. However, due to the small scale of this single molecule reactor, PNRSS is designed as a sensing instead of a preparative method at the moment."

Reviewer #6 (Remarks to the Author):

Reviewer Comments:

The manuscript "Programmable Nano-Reactors for Stochastic Sensing" by Wendong Jia et al, studies several chemical reactions, for up to 20 single molecules, using a protein nanopore. To answer this challenge, the authors designed and manufactured an elegant nanoreactor with a MspA nanopore used to dock a streptavidin tethered PNRSS strand. A part of this strand contains a versatile reaction section with different kinds of reactive sites to perform the chemical reactions. This work's central claim is shown: the ability to use the nanoreactor as a versatile platform to measure the association/dissociation rates and binding constants of different chemical reactions at the single-molecule level. Another interesting claim is the ability to detect chemical intermediates.

All the comments made in the previous review were well answered along with appropriate modifications in the manuscript. Furthermore, I would like to emphasize that new experiments were achieved showing this system can be universally used with protein nanopores, demonstrated here with an alpha-hemolysin pore. This new finding clearly increases the significance of the study elevating it to a substantially valuable contribution to the field. Additionally, a number of 20 molecules were analyzed in this paper while most of the nanopore community usually focuses on just a few molecules. This is an excellent work and this is why I strongly recommend this paper for publication in Nature Communication.

Author Response:

We sincerely acknowledge reviewer 6 for a valuable summary of the core innovations of the newly developed technique PNRSS and for your recognition of our contribution to the field. Your strong recommendation of this paper to be published in Nature Communications is sincerely appreciated.

Reviewer Comments:

Comment to the authors:

"The approach of attaching a streptavidin (or neutravidin...) to the end of the polymer chain (ss DNA or polypeptide) to control the entry of the chimeric molecule and to immobilize this molecule inside the nanopore in order to detect analytes or immobilized chains is not reported in the manuscript introduction. This idea is not new, and the authors should cite previous work involving this kind of system"

From the answer given by the authors I believe the comment was maybe not clear regarding chimeric molecules trapped inside nanopores. Here are some examples of this approach already published (non-exhaustive list):
Pastoriza et al, ACS Nano, 2014 ; Rosen et al, Nat. Biotech, 2014 ; Rodriguez Larrea et al, Nat. Nanotech, 2013 ; Rodriguez Larrea et al, Nat. Comm., 2014 ; Nivala et al, ACS Nano, 2014. ; Nivala et al, Nat. Biotech., 2013

Author Response:

We highly acknowledge reviewer 6 for listing these references. A brief description of these references has been added to the revised manuscript in the revised manuscript to complement the discussions.

"Some previous demonstrations of chimeric molecules trapped inside the pore lumen may be inspiring in the design of a PNRSS strand using peptide components. However, observation of single molecule chemical reactions has not been reported with these trapped chimeric molecules so far."

Reviewer #7 (Remarks to the Author):

Reviewer Comments:

This thorough manuscript revision has increased the strength and readability of this work. The authors included the standard inter-event and dwell time histograms which adequately addresses one of the major concerns raised. Also the newly introduced detection of vitamin B6 in urine significantly strengthens the potential applications for PNRSS. I feel the authors have updated the manuscript in a way that most of reviewer's reservations have been remedied but a few concern can still be rectified.

Author Response:

We highly acknowledge reviewer 7 for your comment that the manuscript has been thoroughly revised and major concerns from previous reviewers have been successfully addressed. The remaining issues have been addressed below in a point by point style.

Reviewer Comments:

The concerns raised about how time constants were determined can still be addressed. The authors should draw attention to Fig. S4 earlier in the "Nanopore measurements and data analysis" section if they want to use this figure as their only explanation of how they are determined.

Author Response:

Thanks reviewer 7 for your suggestion. Reference to Supplementary fig. 4 has been drawn earlier as requested. The first mentioning of Supplementary fig. 4 is pasted below for your references.

"Core parameters that quantitatively describe any PNRSS event are summarized in Supplementary fig. 4, in which the dwell time t_{off} and the inter-event interval t_{on} are defined. The blockage amplitude, ΔI is defined as $I_b - I_p$."

Reviewer Comments:

Also the trace time scales can still be altered to allow the reader to visually confirm the t_{off} and t_{on} values being reported. I understand that the authors are using Fig. 1d as an example trace but it would be helpful if they mentioned the concentration of Ni²⁺. This will help the reader confirm the presented data for themselves. The authors can replace the 500 ms axis with 100 ms in the

highlighted region of the trace so the ~12 ms dwell time events are more obvious.

Author Response:

We acknowledge reviewer 7 for this suggestion. The mentioned display item has been updated with a further zoomed in version. A 100 ms time scale instead of a 500 ms time scale is now applied. Details of event amplitude, event dwell time and inter-event intervals are now more clearly visualized in Fig. 1d.

Reviewer Comments:

The authors strength the paper by introducing material showing PNRSS detection of neutral and positively charge analytes. Both the neutral catechol and positive norepinephrine provide critical data showing that the mean dwell times are independent of voltage. It is not clear why the association of the charged analyte is linearly dependent on the applied voltage. I would expect it to be exponentially dependent based on the equation $k_{on} = A(0) e^{(-q\Delta U/(kBT))}$. The authors should discuss why they observe a linear voltage dependence.

Author Response:

We acknowledge reviewer 7 for the comment. The linear dependence of the association rate on the voltage is due to the contribution of the electrophoretic force. Based on our experimental observation, for catechol, which is electrical neutral, the association rate is almost independent of the applied voltage. However, for norepinephrine, which is electrical positive, the association rate is obviously linearly dependent on the applied voltage. Briefly, a larger electrophoretic force will re-distribute the local concentration of the mobile reactant in the vicinity of the fixed reactant in the pore lumen. A suitable quantitative model to describe this would be a Poisson-Nernst model. This may be carried out in a follow up study on the topic.

The mentioned equation, which is the Arrhenius equation $k_{on} = A(0) e^{(-E_a/(kBT))}$ actually better describes why the reaction rate is exponentially related to the measurement temperature (Supplementary Fig 57). The E_a describes the activation energy and is independent of the applied voltage across a membrane.

Reviewer Comments:

This version of the manuscript better assists the reader to supplementary information which helps address a lot of the previous concerns. Therefore, I

feel the authors have satisfactorily addressed the concerns raised by reviewer # 4.

Author Response:

We thank reviewer 7 again for your support of this manuscript. Your valuable suggestions and comments are also extremely appreciated.